## RESEARCH ARTICLE

# Bidirectional interaction between protocadherin 8 and the transcription factor Dbx1 regulates cerebral cortex development

Andrzej W. Cwetsch[1,2,3,‡], Sofia Ferreira[1,2], Elodie Delberghe[1,2,*], Javier Gilabert-Juan[4,*], Matthieu X. Moreau[1,2], Yoann Saillour[1,2], Pau García-Bolufer[3], Saray Calvo-Parra[3], Jose González-Martínez[5], Durcia Massoukou[1,2], Ugo Borello[6], Frédéric Causeret[1,2] and Alessandra Pierani[1,2,7,‡]

## ABSTRACT

Brain development requires correct tissue patterning and production of appropriate cell types. Transcription factors play essential roles in these processes, regulating the expression of target genes responsible for the specific features of neuronal subtypes. Cell adhesion molecules are key components of developmental processes that control cell sorting, migration, neurite outgrowth/guidance and synaptogenesis. To date, the link between transcription factors and cell adhesion molecules has been considered unidirectional. Here, we demonstrate that ectopic expression of Dbx1 leads to spatiotemporally restricted increased expression of *Pcdh8* and cell aggregation, together with changes in neuronal identity. Surprisingly, ectopic Pcdh8 expression also induces Dbx1 expression, as well as a complete reorganisation of apico-basal polarity and dorso-ventral patterning via Notch signalling. Altogether, our work therefore points to cell adhesion molecules as unexpected, yet important, players in the regulation of cell identity and, in particular, Pcdh8 through its bidirectional interaction with the Dbx1 transcription factor.

KEY WORDS: Pcdh8, Dbx1, Cortical development, Neuronal identity, Adhesion molecules, Transcription factors

## INTRODUCTION

Early steps of brain development rely on the tightly controlled action of transcription factors (TFs) that organise the dorso-ventral (DV) axis of the central nervous system (CNS). The precise expression of TFs in domains along the DV axis ensures robust patterning, and

[1]Université Paris Cité, Institute of Psychiatry and Neuroscience of Paris (IPNP), INSERM U1266, Team Genetics and Development of the Cerebral Cortex, 75014 Paris, France. [2]Université Paris Cité, Imagine Institute, Team Genetics and Development of the Cerebral, Cortex, F-75015 Paris, France. [3]Departamento de Biología Celular, Biología Funcional y Antropología Física, Instituto de Biotecnología y Biomedicina (BIOTECMED), Universidad de Valencia, 46100 Burjassot, Spain. [4]Departamento de Anatomía, Histología y Neurociencia, Universidad Autónoma de Madrid, 28039 Madrid, Spain. [5]Cell Division and Cancer group, Spanish National Cancer Research Centre (CNIO), 28029 Madrid, Spain. [6]Unit of Cell and Developmental Biology, Department of Biology, University of Pisa, 56127 Pisa, Italy. [7]GHU-Paris Psychiatrie et Neurosciences, Hôpital Sainte Anne, F-75014 Paris, France.
*These authors contributed equally to this work

‡Authors for correspondence (alessandra.pierani@inserm.fr; andrzej.cwetsch@uv.es)

A.W.C., 0000-0002-8156-1218; S.F., 0000-0003-2066-6865; J.G.-J., 0000-0001-5187-3942; M.X.M., 0000-0002-2592-2373; Y.S., 0000-0002-5110-9239; P.G.-B., 0000-0001-6553-5696; J.G.-M., 0000-0002-3850-6803; U.B., 0000-0001-9812-579X; F.C., 0000-0002-0543-4938; A.P., 0000-0002-4872-4791

their faulty regulation leads to the disruption of the DV organisation and/or cell identity (Arai et al., 2019; Godbole et al., 2017; Leung et al., 2022; Marklund et al., 2010; Pierani et al., 1999). Among these TFs, developing brain homeobox 1 (Dbx1) acts as a potent DV patterning and cell-fate determinant in different regions of the CNS, such as the mouse spinal cord (Pierani et al., 2001), rhombencephalon (Bouvier et al., 2010) and diencephalon (Sokolowski et al., 2016). In the mouse developing telencephalon, Dbx1 exhibits a very precise expression in ventral pallium (VP) progenitors just above the pallial-subpallial boundary (PSB) and the septum (Bielle et al., 2005). Recently, it was shown that its ectopic expression in the dorsal developing cerebral cortex (Arai et al., 2019) promotes Cajal-Retzius and subplate-like cell fate (Arai et al., 2019), with subplate identity depending on primate-specific *cis*-acting elements acting in postmitotic neurons. In the spinal cord, Dbx1 has been shown to regulate DV patterning expression profiles of TFs and to establish subtype-specific interneuron fate (Pierani et al., 2001). This is mediated by controlling the DV expression profile of Notch ligands, namely suppressing Jag1 expression and promoting Delta1 (Marklund et al., 2010). Last, Dbx1 was reported to drive cell cycle exit in birds and mice (Arai et al., 2019; García-Moreno et al., 2018; Nomura et al., 2008). However, how Dbx1 influences cell fate at the molecular level in the developing cerebral cortex remains unknown (Leung et al., 2022).

Protocadherins are cadherin-superfamily adhesion molecules classified as clustered (cPcdhs) and non-clustered Pcdhs. Their expression is regulated in a spatiotemporal manner during development, being involved in multiple processes of CNS formation. The non-clustered δ-Pcdhs (i.e. Pcdh8, Pcdh9 and Pcdh19) have been associated with dendrite and axon formation, morphology, migration and guidance (Bassani et al., 2018; Biswas et al., 2010; Bruining et al., 2015; Cooper et al., 2015; Cwetsch et al., 2022; Pancho et al., 2020; Yasuda et al., 2007), and human mutations associate with several neurodevelopmental disorders, such as autism, schizophrenia and epilepsy (Kahr et al., 2013). In the mammalian neocortex, patterned Pcdh expression biases the fine spatial and functional organisation of individual excitatory neurons (Lv et al., 2022). However, whether Pcdhs play a role in regulating TF expression to establish neuronal identities or DV patterning in embryonic murine brain is unknown.

We previously demonstrated that ectopic expression (EE) of Dbx1 (hereafter Dbx1 EE) at embryonic day (E) 11.5 in the dorsal pallium (DP) of the developing mouse brain imparts cell identity by inducing ectopic Cajal-Retzius and subplate-like neuron fates (Arai et al., 2019), thus resulting in changes in cell fate in the DP. The molecular mechanism behind this process remains, however, unknown. Here, we report that Dbx1 EE in lateral/ventral pallium (LP/VP) progenitor domains induces cell aggregation through Pcdh8 upregulation and that Pcdh8 is required for Dbx1-driven neuronal fate specification,

particularly the generation of Nr4a2$^+$ claustro-amygdalar complex (CLA)-like neurons. We used single-cell RNA sequencing (scRNAseq) to characterise distinct expression dynamics for both genes during neuronal differentiation at the VP and septum. Consistently, we reveal that Pcdh8 EE induced a cell-autonomous upregulation of Dbx1 as well as non-cell-autonomous defects in cell proliferation and tissue patterning, resulting in a massive reorganisation of the cortical primordium. Finally, we identified the Notch ligand Jag1 as a downstream effector of Pcdh8. Our findings therefore point to cell adhesion molecules, particularly Pcdh8, as players acting not only downstream, but also upstream of TFs during brain development.

## RESULTS
### Ectopic Dbx1 expression in the LP/VP induces cell aggregation
Using *in utero* electroporation (IUE), we previously investigated the effects of ectopic pCAGGS-Dbx1-iresGFP expression in the E11.5 mouse DP. We found that Dbx1 EE induced the production of neurons harbouring features of Cajal-Retzius [Reln$^+$/Calr$^+$ (calretinin; also known as Calb2)] and subplate-like Nr4a2$^+$Bcl11b$^+$ (also known as Nurr1$^+$Ctip2$^+$) fates 48 h after electroporation (Arai et al., 2019). In the present study, we first validated our experiments by confirming that all Dbx1 EE-electroporated GFP$^+$ cells expressed Dbx1 (Fig. S1A-C). Surprisingly, when electroporation of Dbx1 at E11.5 encompassed LP/VP progenitor domains, we noticed 48 h later that transfected cells tended to aggregate, forming cohesive clusters (Fig. 1A,B; Fig. S1B,D,E). Such clusters were preferentially formed in the LP (Fig. 1C,D) at the border between the ventricular/subventricular zone (VZ/SVZ) and the cortical plate (CP) (Fig. 1B; Fig. S1B). They were rarely observed in the medial pallium (MP) or DP, and in GFP-electroporated control brains (Fig. 1B-D). Aggregation also failed to occur when IUE was performed 1 day later, at E12.5 (Fig. S1A,F), suggesting a spatio-temporal modulation in the response to Dbx1 EE. Knowing that cell aggregation often relies on stronger cell-cell or cell-substrate adhesion, we next investigated whether Dbx1 EE affected cell adhesiveness. To this end, we performed a well-established adhesion assay, as previously described (Porlan et al., 2014). Neuronal stem cells were transfected *in vitro* with Dbx1 EE (iresGFP) or a control vector (iresGFP) and, after 24 h, GFP$^+$ cells were seeded onto NC929 cell monolayers (overexpressing Ncad; also known as Cdh2), and the number of attached cells was then quantified (Fig. S1G). The number of attached GFP$^+$ cells, normalised to transfection efficiency [total GFP$^+$ cells before seeding quantified by fluorescence-activated cell sorting (FACS)], showed that Dbx1 EE significantly enhanced adhesion properties (Fig. S1H,I).

### Ectopic Dbx1 expression in the lateral/ventral pallium induces Nr4a2$^+$ neurons resembling CLA identity
We then investigated whether Dbx1 EE and increased adhesion influenced cell fate. Consistent with our previous findings, we found numerous Nr4a2-expressing cells within Dbx1-induced aggregates, but absent in control GFP-electroporated cells (Fig. 1B,E,G; Fig. S2A). These clusters, however, lacked Ctip2 expression, a marker of early-born and subplate neurons, unlike what is observed following IUE of the DP (Arai et al., 2019). Instead, Ctip2 expression appeared to be excluded from GFP$^+$ regions (Fig. 1F,H; Fig. S2B). Quantitative analysis of immunofluorescence (IF) intensity of Nr4a2$^+$ or Ctip2$^+$ neurons aligned with the GFP signal of Dbx1 EE cells confirmed a positive correlation with Nr4a2 but a negative correlation with Ctip2 (Fig. 1G,H). Surprisingly, Dbx1-induced aggregates were also Reln$^-$ and Calb2$^-$, with some individual Reln$^+$ and Calb2$^+$ cells

at the cortical surface (Fig. S2C,D). Moreover, the fraction of postmitotic pallial GFP$^+$Bhlhe22$^+$ was unchanged between Dbx1 EE and control conditions (Fig. S2E,F), indicating that their fate it is not specifically induced by Dbx1 EE in the LP/VP. To determine whether Dbx1-induced neurons persist and acquire a defined identity at later stages, we analysed embryos 7 days after IUE (E11.5-E18.5). At this stage, Dbx1 EE$^+$/Nr4a2$^+$ cells accumulated in the intermediate zone (IZ)/subplate (Fig. 1Ia,c). Within the CP, Dbx1 EE$^+$ were Nr4a2$^-$ and coexisted with Nr4a2$^+$ non-electroporated cells (Fig. 1Ib,d), the latter being endogenous late-born glutamatergic neurons of the lateral cortex (Fang et al., 2021). To assess whether Nr4a2$^+$ neurons derive from Dbx1-expressing progenitors in the developing cerebral cortex, we used *Dbx1$^{nlslacZ/+}$* mice (in which the *lacZ* gene reporter encoding β-galactosidase (β-gal) is driven by the *Dbx1* promoter), allowing short-term lineage tracing of Dbx1-derived cells. We confirmed that β-gal$^+$ Dbx1-derived cells located in the postmitotic compartment of the VP expressed Nr4a2 at E12.5 (Fig. S3A). Furthermore, we used *Dbx1$^{Cre}$;Rosa26$^{YFP}$* animals to permanently trace Dbx1-derived progeny. In the rostral LP at E12.5 and later at E14.5, no YFP$^+$Nr4a2$^+$ cells were detected in the subplate (Fig. S3B,C), consistent with previous observations at E16.5 (Arai et al., 2019). This shows that Dbx1 progenitors do not give rise to Nr4a2$^+$ subplate neurons in the mouse. Instead, in intermediate rostro-caudal sections at the VP, we observed ventrally located groups of YFP$^+$Nr4a2$^+$ cells both at E12.5 and E14.5 (Fig. S3Bb,Bc,Cc). Together with reported Dbx1-lineage tracing data (Puelles et al., 2016), these findings indicate that Dbx1 progenitors in the mouse VP originate pallial Nr4a2$^+$ neurons that migrate ventrally, likely corresponding to future CLA neurons (Puelles et al., 2016), and not subplate neurons.

Altogether, these results show that Dbx1 EE promotes cell-cell adhesion in a cell-autonomous manner within the LP/VP, leading to the induction of Nr4a2$^+$Ctip2$^-$ cells resembling CLA-like neurons. In addition, Dbx1-induced cell aggregation is spatiotemporally restricted by the competence of the cortical neuroepithelium, with no aggregation in DP (Arai et al., 2019), correlating with Dbx1 EE promoting exclusively Cajal-Retzius and subplate-like fates at E11.5 (Arai et al., 2019), and aggregation in LP/VP with CLA-like fate induction. Moreover, we confirm that endogenous VP Dbx1$^+$ progenitors do not generate subplate neurons in mice, and subplate-like fate arises from primate-like Dbx1 expression in postmitotic neurons (Arai et al., 2019), whereas Cajal-Retzius and CLA-like neurons are endogenous fates driven by VP Dbx1$^+$ progenitors in mice.

### Pcdh8 and Dbx1 show concomitant expression in pallial progenitors and Dbx1-derived neurons
To begin investigating the molecular mechanisms behind Dbx1-induced aggregation, we took advantage of previously published bulk transcriptomic profiles of FACS-purified Dbx1-derived cells from dorsomedial and dorsolateral cortices of E12.5 *Dbx1$^{Cre}$;Rosa26$^{YFP}$* embryos (Fig. 2A-C) (Griveau et al., 2010). We compared the expression level of genes belonging to the Protocadherin (Fig. 2B) and Cadherin (Fig. 2C) families, two major players in neuronal adhesion (Halbleib and Nelson, 2006; Peek et al., 2017; Suzuki and Takeichi, 2008). We found that *Ncad*, *Pcdh8*, *Pcdh9* and *Pcdh19* stood out, with ~10-fold higher expression levels in comparison to other adhesion molecules (Fig. 2B,C). To better appreciate their possible implication downstream of Dbx1, we used scRNAseq. We first used a previously generated dataset sampling cell diversity around the PSB at E12.5 (Moreau et al., 2021) (Fig. 2D,E). We found *Ncad*, *Pcdh9* and *Pcdh19* broadly expressed without specific

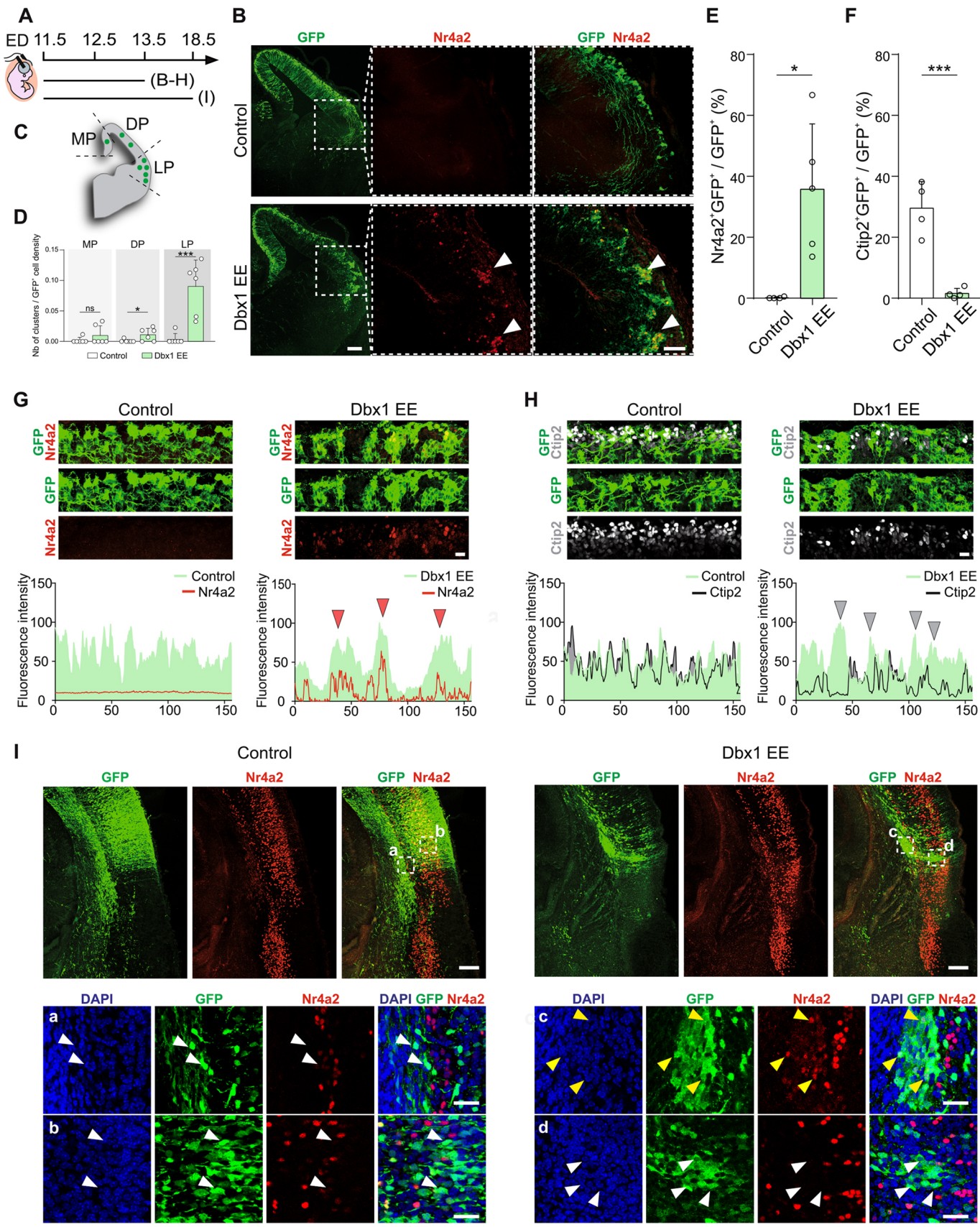

**Fig. 1.** See next page for legend.

**Fig. 1. Dbx1 EE induces cell aggregation in the lateral/ventral pallium.**
(A) Timeline of the *in utero* electroporation (IUE). ED, electroporation day. (B) Confocal images of GFP (green) in coronal sections of E13.5 mouse brain cortices electroporated at E11.5 with control or Dbx1 EE, co-labelled with Nr4a2 (red). Dashed squares magnified on the right. White arrowheads indicate aggregates. (C) Aggregate analysis scheme in the medial pallium (MP), dorsal pallium (DP) and lateral pallium (LP). (D) Quantification of GFP$^+$ cluster formation across the pallium. Circles represent individual section values (*n*=3 electroporations/condition, two sections/mouse). One-way ANOVA, post hoc Holm–Šidák: ns, not significant, *P*=0.0399, ***P*=0.0007. (E,F) Percentages of Nr4a2$^+$GFP$^+$ (E) and Ctip2$^+$GFP$^+$ (F) cells among GFP$^+$ cells. Circles represent values from independent electroporated embryos [*n*=5 (E); *n*=4 (F)]. Two-tailed unpaired Student's *t*-test, (E) *P*=0.01, (F) ***P*=0.0008. (G,H) Top: confocal images of E13.5 mouse coronal sections showing GFP$^+$ (green) cells electroporated with control or Dbx1 EE, co-labelled with Nr4a2 (red, G) or Ctip2, grey) (H). Bottom: fluorescent intensities of GFP$^+$ cells (green area) and Nr4a2$^+$ (red line, G) or Ctip2$^+$ (grey line, H), along the cortical surface (µm). Red (G) or grey (H) arrowheads show positive (Dbx1 EE GFP$^+$ areas and Nr4a2 labelling) or negative (Dbx1 EE GFP$^+$ areas and Ctip2 labelling) fluorescence correlation. (I) Top: confocal images of GFP (green) in coronal sections of E18.5 mouse brain cortices electroporated at E11.5 with control or Dbx1 EE, co-labelled with Nr4a2 (red) and DAPI (blue). Bottom: dashed squares magnified below (Ia-Id). White and yellow arrowheads indicate Dbx1 EE$^-$/Nr4a2$^+$ and Dbx1 EE$^+$/Nr4a2$^+$ cells in the electroporated area. Data are mean±s.e.m. Scale bars: 200 µm (B,I); 100 µm (magnification in B); 25 µm (G, magnification in I).

enrichment (https://apps.institutimagine.org/mouse_pallium/), which was confirmed by *in situ* hybridisation (ISH) (Fig. S4A-C). In contrast, we detected *Pcdh8* expression in apical progenitors (APs) of the subpallium, VP and septum, but not of LP and DP regions (Fig. 2D,E; Fig. S4D,E). Since *Dbx1* expression initiates in APs of the VP and strongly increases during the transition to the basal progenitor (BP) state (Fig. 2E), we compared the temporal dynamics of *Dbx1* and *Pcdh8* along a previously reconstructed VP differentiation trajectory, which orders cells by pseudotime from APs to neurons at E12.5 (Moreau et al., 2021) (Fig. 2E). *Pcdh8* and *Dbx1* are expressed sequentially in the VP lineage, with *Pcdh8* peaking in APs, followed by *Dbx1* in BPs and *Nr4a2* in neurons (Fig. 2E). To further investigate their co-expression at earlier stages in the VP, we produced an scRNAseq dataset from the entire telencephalic vesicle at E11.5-E12 and detected *Dbx1*$^+$*Pcdh8*$^+$ cells at the AP-to-BP transition (Fig. 2F), likely belonging to the VP lineage, as *Dbx1* expression is confined to progenitors in this region (Fig. 2G) (Moreau et al., 2021). Consistently, *Pcdh8* mRNA and protein were enriched in progenitor domains adjacent to the most characteristic *Dbx1*-expressing regions of the telencephalon, specifically the VP dorsal to the PSB, and the septum, in embryonic wild-type (WT) brain sections (Bielle et al., 2005) (Fig. 2G). Together, these observations suggest a possible, albeit transient, direct interaction between Dbx1 and Pcdh8 in the VP. We also performed scRNAseq on septal explants collected from E11.5 *PGK*$^{Cre}$;*Rosa26*$^{YFP}$ and E12.5 *Dbx1*$^{Cre}$; *Rosa26*$^{Tomato}$ embryos, the latter enabling genetic tracing of Dbx1-derived cells (Fig. S4D,E) (https://apps.institutimagine.org/mouse_septum/). We observed a robust co-expression of *Dbx1* and *Pcdh8* in both progenitors and septal-derived postmitotic excitatory neurons (*Tbr1*$^-$/*Slc17a6*$^+$) (Fig. S4D,E). We further validated the co-expression of *Dbx1* and *Pcdh8* by fluorescent ISH (FISH) at E12.5 (Fig. 2H) by observing co-expressing cells within the postmitotic compartment of the septum (Fig. 2H1a, 1b) in a ventrally migrating stream of postmitotic Dbx1-derived cells (Bielle et al., 2005; Griveau et al., 2010) (Fig. 2H2c). Additionally, we detected co-expression in progenitors at the VP/PSB (Fig. 2H3d). At the protein level, Pcdh8 expression in the VP and in Dbx1-derived neurons was confirmed by

co-immunostaining of β-gal and Pcdh8 (antibody validation at E12.5, Fig. 2G) on brain sections of E11.5 and E12.5 *Dbx1*$^{lacZ/+}$ embryos (Fig. S5A,B). In the postmitotic compartment of the VP of E12.5 WT animals, we found Pcdh8$^+$Nr4a2$^+$ co-expressing neurons (Fig. S5C). Finally, to confirm the epistasis between Dbx1 and Pcdh8, we analysed *Pcdh8* expression in E11.5 *Dbx1*$^{lacZ/lacZ}$ knockout (KO) animals. We observed altered expression domains in the septum and VP/PSB progenitor regions and, importantly, a reduction in ventral and lateral postmitotic compartments where *Pcdh8*$^+$ cells are normally present (Fig. S5D).

We therefore conclude that *Dbx1* and *Pcdh8* display interrelated dynamic expression patterns in the developing telencephalon, occurring both concomitantly in AP/BP progenitors between E11.5 and E12.5, and sequentially during neuronal differentiation in both the VP and septal lineages, suggesting possible direct interaction. Furthermore, co-expression studies combined with tracing analysis confirm the existence of endogenous Dbx1$^+$Pcdh8$^+$ and Dbx1-derived Nr4a2$^+$Pcdh8$^+$ progenitors and neurons in the embryonic telencephalon.

**Dbx1 EE induces Pcdh8 expression**
To address whether Dbx1 EE regulates *Pcdh8* expression, we performed quantitative reverse transcription polymerase chain reaction (RT-qPCR) on lateral cortical tissue dissected 48 h post-electroporation (E11.5-E13.5) with Dbx1-iresGFP or GFP control expression vectors (Fig. 3A,B). We included in the analysis *Reln*, *Calb2* and *Nr4a2* given their known upregulation upon Dbx1 EE in the DP (Arai et al., 2019), *Ctip2* given its exclusion from Dbx1-induced aggregates (Fig. 1F,H; Fig. S2B) and *Pcdh9*, *Pcdh19* and *Ncad*. As expected, *Dbx1* mRNA levels increased in Dbx1-electroporated regions compared to controls (Fig. 3C). The expression levels of *Pcdh8*, *Nr4a2* and *Calb2* transcripts were significantly increased in the electroporated areas, while the remaining genes remained unchanged (Fig. 3D). Similar results were obtained 24 h post-electroporation (E11.5-E12.5), with a significant upregulation of *Dbx1*, *Pcdh8* and *Nr4a2* transcripts, arguing for a rapid, if not direct, regulatory effect (Fig. 3E,F). Additionally, following Dbx1 EE (E11.5-E13.5), we observed a significant negative correlation between *Pcdh19* and *Dbx1* gene expression (Fig. S6A), such that samples with higher *Dbx1* expression exhibited lower *Pcdh19* levels, pointing to a *Dbx1*-dose dependent regulation and other possible partners in gene co-regulation networks. This relationship was not observed for other cadherins (i.e. *Pcdh8*, *Pcdh9* and *Ncad*). Consistently, *Pcdh19* expression on sections was detected in DP but not in LP/VP progenitors at E12.5-E13.5 (Fig. S4B,C). In contrast, Dbx1 IUE at E12.5 (E12.5-E14.5) no longer altered *Pcdh8* and *Nr4a2* expression, but resulted in the upregulation of *Pcdh9*, *Pcdh19*, *Ncad* and *Ctip2* (Fig. S6B-E), which overall aligns with the absence of cell aggregates upon Dbx1 EE at this developmental stage (Fig. S6F).

To validate these findings, and determine whether changes in gene expression are cell-autonomous, we performed co-immunostaining of Pcdh8 and Dbx1 on E11.5 Dbx1-electroporated brain sections. We confirmed higher cell-autonomous Pcdh8 protein expression in Dbx1 EE cells at E13.5 (Fig. 3G). At E18.5, the picture was more complex, as we observed both clusters of Pcdh8$^{high}$Ctip2$^-$Nr4a2$^+$ cells below the CP (Fig. S7Aa,c; Fig. 1Ia,c), and Pcdh8$^{low}$Ctip2$^+$Nr4a2$^-$ cell streams within the CP (Fig. S7Ab,d; Fig. 1Ib,d). Furthermore, following IUE at E11.5, the regular lateral Pcdh19 expression on E13.5 control sections was strongly reduced specifically where Dbx1-induced aggregates were formed (Fig. S7B), consistent with the negative correlation with *Dbx1* gene expression (Fig. S6A),

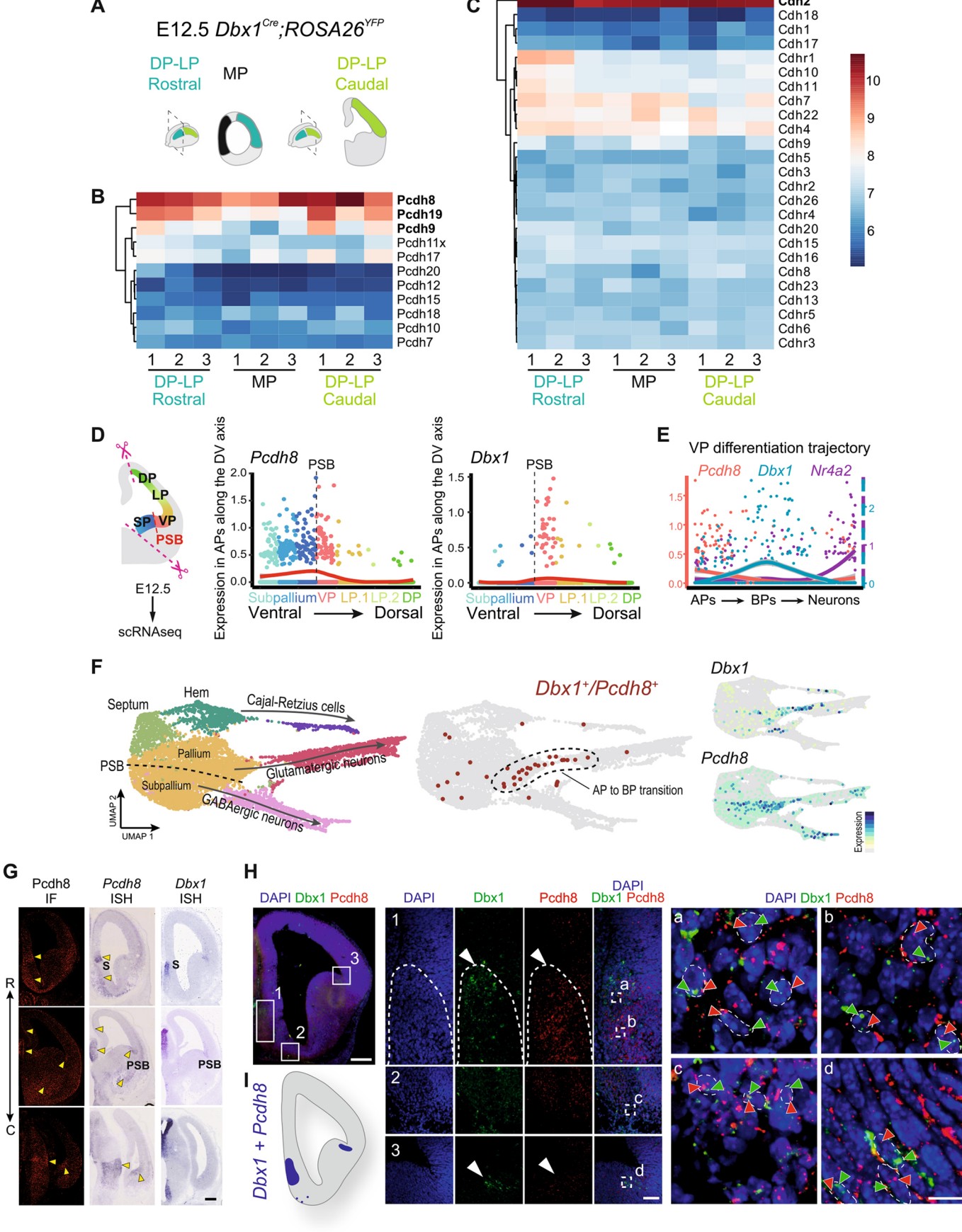

**Fig. 2.** See next page for legend.

**Fig. 2. The endogenous expression of Pcdh8 and Dbx1 partially overlaps.** (A) Experimental design: E12.5 *Dbx1^Cre^;Rosa26^YFP^* brain showing dissected regions of medial pallium (MP), dorsal pallium (DP) and lateral pallium (LP) at rostral and caudal levels used for bulk transcription profiling of FACS-sorted YFP⁺ cells. (B,C) Heatmaps of microarray analysis from E12.5 *Dbx1^Cre^;Rosa26^YFP^* brains (*n*=3) showing expression levels of Pcdhs (B) and cadherins (Cdhs) (C). The colour scale (right) represents expression intensity from low (blue) to high (red). (D) Left: schematic of the dissected brain region from E12.5 WT used for scRNAseq. Right: comparison of *Pcdh8* and *Dbx1* gene expression trends along pseudo-DV scores. Each dot represents a single apical progenitor (AP), color-coded by domain. SP, subpallium; VP, ventral pallium; PSB, pallial-subpallial boundary. The red curve indicates the smoothed expression profile. (E) Comparison of *Pcdh8*, *Dbx1* and *Nr4a2* gene expression trends along the VP differentiation trajectory, from APs to basal progenitors (BPs) to neurons, mapped onto pseudotime. (F) Left: UMAP of the scRNAseq dataset from E11.5-E12 WT telencephalic vesicles, with cells colour-coded by type. Neuron types are indicated, and arrows represent the main differentiation trajectories. The dashed line marks the PSB. Middle: Cells co-expressing *Dbx1* and *Pcdh8* (dark red) are mostly found at the AP-to-BP transition (dashed area). Right: UMAP visualisation of *Dbx1* and *Pcdh8* gene expression. (G) Left: confocal images of Pcdh8 (red) immunofluorescence (IF) in coronal sections of E12.5 WT mouse brain. Right: bright-field images of *Pcdh8* and *Dbx1* ISH in E12.5 WT coronal sections. The double arrow line (far left) indicates the rostro-caudal (R-C) axis. Yellow arrowheads indicate strong correspondence between Pcdh8 signals in IF and ISH, validating the anti-Pcdh8 serum. S, septum. (H) Confocal images of E12.5 WT mouse coronal brain sections showing *Dbx1* (green) and *Pcdh8* (red) FISH, with DAPI (blue). Regions 1-3: (1) medio-ventral septal postmitotic compartment; (2) postmitotic compartment of pallial cells migrating ventrally; (3) PSB. White arrowheads indicate areas of highest Dbx1 and Pcdh8 expression. Dashed squares (Ha-Hd) magnified on the right; dashed circles outline cells co-expressing *Dbx1* (green arrowheads) and *Pcdh8* (red arrowheads) signals. (I) Schematic showing overlapping regions with high *Dbx1* and *Pcdh8* expression detected by FISH. Scale bars: 100 µm (G and H, upper left); 25 µm (H1-H3); 5 µm (Ha-Hd).

supporting a *Dbx1*-mediated repression of *Pcdh19* transcription. Overall, these findings indicate a rapid and possibly time-restricted Dbx1 EE action on Pcdh8 upregulation as well as Pcdh19 repression in a cell-autonomous manner.

Since aggregation and cell fate allocation differed between DP and LP/VP, and δ-Pcdh (i.e. Pcdh8, Pcdh9 and Pcdh19) mediate exclusively homophilic aggregation while avoiding heterotypic aggregation (Bisogni et al., 2018), we examined the *Pcdh8*, *Pcdh9* and *Pcdh19* ISH data to directly compare their expression patterns at E11.5, E12.5 and E13.5 (Fig. S4A-C). This comparison revealed clear differences between LP and DP expression domains. For example, at E11.5/E12, *Pcdh9* exhibits strong expression in the postmitotic compartment with a lateral^high^-to-medial^low^ gradient, and is highly expressed in progenitors around the PSB, whereas at E12.5, *Pcdh19* is highly expressed in progenitors following a medial^high^-to-lateral^low^ gradient, with almost no expression at E12.5-E13.5, except in postmitotic neurons of the future piriform cortex (Fig. S4A-C). Moreover, expression patterns for all three Pcdhs were extremely dynamic between E11.5 (IUE time point) and E13.5 (analysis time point). These results thus reflect important spatio-temporal changes in δ-Pcdh expression that may influence the responsiveness of the cortical neuroepithelium to Dbx1 between E11.5 and E12.5, pointing to possible distinct Pcdh- and Cdh-related programmes induced by Dbx1 at these developmental stages.

## Pcdh8 is required for Dbx1-induced adhesion and fate specification

To determine the contribution of Pcdh8 to Dbx1-induced cell aggregation, we further investigated the epistasis between Dbx1 and

Pcdh8 by combining Dbx1 EE with *Pcdh8* knockdown (KD). We first designed three shRNA constructs targeting *Pcdh8* and determined their efficiency by western blot analysis 48 h after co-transfection with a *Pcdh8* expression vector in HEK293T cells. We selected the most effective shRNA, which reduced *Pcdh8* expression by 51% compared to a control shRNA (Fig. S8A,B). When electroporated alone, the Pcdh8 shRNA did not induce obvious changes in cell aggregation (Fig. S8D) or cell identity (Nr4a2, Ctip2) (Fig. S8E,F). Next, we co-electroporated E11.5 mouse brains with Dbx1 EE and Pcdh8 shRNA or control shRNA constructs (Fig. 4A-C). Interestingly, we observed no aggregate formation with *Pcdh8* KD (Fig. 4B,C; Fig. S8G), thus indicating that Pcdh8 is required for Dbx1-induced adhesion. Moreover, we observed a reduced number of GFP⁺Nr4a2⁺ cells (Fig. 4D), and an increased number of GFP⁺Ctip2⁺ cells (Fig. 4E) compared to Dbx1 EE+control shRNA or Dbx1 EE alone, indicating that Pcdh8 is also necessary for Dbx1-induced fate determination. Since *Pcdh8* KD rescued the phenotypes induced by Dbx1 EE, we decided to monitor Dbx1 expression upon *Pcdh8* KD. Surprisingly, we observed a marked reduction of GFP⁺Dbx1⁺ cells compared with both Dbx1 EE+control shRNA and Dbx1 EE alone (Fig. 4C,F,G). We thus conclude that Pcdh8 is an essential component in Dbx1-induced aggregation and cell-fate determination.

## Pcdh8 EE results in major reorganisation of pallial domains

To further investigate the epistasis between Dbx1 and Pcdh8, we performed Pcdh8 gain-of-function (Pcdh8 EE) in the LP, which is a territory devoid of Dbx1 expression, by IUE at E11.5 (E11.5-E13.5) using a pCAGGS-Pcdh8-iresGFP plasmid. First, we confirmed Pcdh8 ectopic expression through western blotting (Fig. S8A,C) and IF (Fig. S8H). In addition, ISH revealed similar expression levels of *Dbx1* and *Pcdh8* in both Dbx1 EE and Pcdh8 EE experiments (Fig. S8I). Upon Pcdh8 EE, the electroporated cortical region exhibited increased thickness compared to the contralateral non-electroporated hemisphere (Fig. 5A-C). ISH experiments revealed major tissue reorganisation, consisting in the formation of large circumferential GFP⁺ structures that recapitulated the DV partitioning of pallial domains, with patches or stripes of cells expressing the ventral markers *Shh* or *Gsx2* segregated from cells positive for the dorsal markers *Lhx2* or *Bhlhe22* (Fig. 5D; Fig. S8J, K). Moreover, we observed that markers of cycling progenitors such as Pax6 (Fig. 5E,F) or *Notch1* (Fig. 5D; Fig. S8K) were affected and extended outside of the VZ. Strikingly, *Dbx1*, which is normally restricted mostly to BPs of the VP, was found expressed throughout the electroporated area (Fig. 5D; Fig. S8K). Altogether, Pcdh8 EE caused the local overgrowth of rosette-like structures, characterised by an outer layer of *Pcdh8*, *Dbx1*, *Bhlhe22* and *Shh*-expressing cells, segregating and organising *Pax6*, *Lhx2* and *Notch1*-positive cells at the centre. The rosettes often displayed a central lumen lined by PH3⁺ mitotic cells, surrounded by GFP⁺ (Pcdh8⁺) cells (Fig. 5G, H) and delineated by Ncad⁺ apical junctions (Fig. 5I). We observed the delamination of PH3⁺ APs (Fig. 5G) that correlated with the disruption of Ncad⁺ adherens junctions in the neuroepithelial cells lining the ventricle (Fig. 5I).

We therefore conclude that ectopic Pcdh8 expression disrupts the local organisation of the telencephalic neuroepithelium through the modulation of cell adhesion, resulting in alterations in DV patterning gene expression and apico-basal polarity within the LP.

## Cell-autonomous control of cell fate by Pcdh8

The very precise segregation of gene expression in outer and central domains of the observed rosette-like structures prompted us to

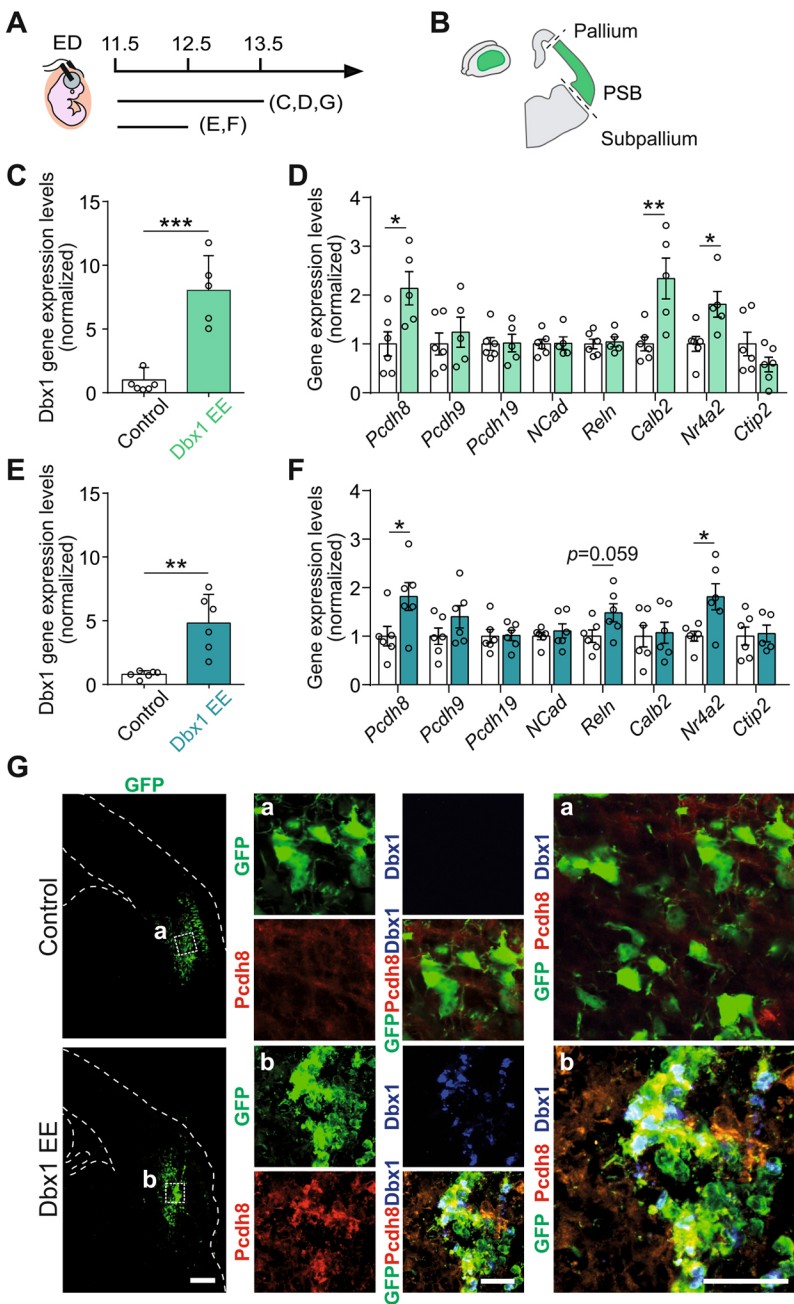

**Fig. 3. Dbx1 EE induces *Pcdh8* gene expression in the electroporated area.** (A) Timeline of the *in utero* electroporation (IUE). ED, electroporation day. (B) Electroporated region dissection scheme. (C-F) qPCR quantifications of (C,E) *Dbx1* or (D,F) *Pcdh8*, *Pcdh9*, *Pcdh19*, *Ncad*, *Reln*, *Calb2*, *Nr4a2* and *Ctip2* gene expression upon IUE at E11.5 of a control vector or Dbx1 EE and analysis at E13.5 (C,D) and E12.5 (E,F). Gene expression levels normalised internally to *Gapdh* expression. Data are mean ±s.e.m.; circles represent values from independent electroporated embryos (*n*=6 controls, *n*=5-6 Dbx1 EE). Two-tailed unpaired Student's *t*-test: (C) ***$P$=0.0002; (E) **$P$=0.0014; (D,F) *$P \le 0.05$ and **$P \le 0.01$. (G) Confocal images of GFP (green) in coronal sections of E13.5 mouse brains electroporated at E11.5 with control or Dbx1 EE, co-labelled with Pcdh8 (red) and Dbx1 (blue). Dashed squares (Ga,Gb) magnified on the right. Dashed lines outline the pallium and ventricle. Scale bars: 200 µm (G); 100 µm (magnification in Ga, Gb).

evaluate more precisely the cell-autonomous versus non-cell-autonomous function of Pcdh8. We initially performed Dbx1 IF on brain sections of E11.5-electroporated embryos, 48 h after IUE with Pcdh8 EE (E11.5-E13.5). The analysis revealed extensive co-localisation, with 90%±5% of Pcdh8 EE-expressing cells (GFP[+]) being Dbx1[+], compared to 0% of control GFP-transfected cells (Fig. 6A-C). Furthermore, 16.47%±2.38% of Pcdh8 EE-expressing cells were Nr4a2[+]Ctip2[−], compared to only 0.29% in controls (Fig. 6D,E), while the Nr4a2[+]Ctip2[+] subplate-like cells remained low and comparable between Pcdh8 EE and control conditions (Fig. 6D, F). This suggests that, similar to Dbx1 EE, Pcdh8 EE does not produce subplate neurons in the LP. On the other hand, the proportion of Ctip2[+]Nr4a2[−] cells was reduced among Pcdh8-transfected cells (19.92±0.95%) compared to controls (29.54±4.37%) (Fig. S9A,B). Additionally, and in contrast to Dbx1 EE, the proportion of pallial Bhlhe22-expressing neurons was markedly increased in Pcdh8-

transfected cells (42.3±0.95%) relative to controls (16.7%±11.8%), while *Pcdh8* KD did not alter this cell fate (Fig. S9C,D). These findings are consistent with the cell fate effects of Dbx1 EE (Fig. 1), indicating that Pcdh8 can regulate a similar cell fate, likely of the CLA lineage (Nr4a2[+]Ctip2[−]), in the developing LP/VP in a cell-autonomous manner.

## Pcdh8 EE controls progenitor identity by impairing Jag1-mediated Notch signalling

To further investigate the mechanisms by which Pcdh8 controls the establishment of neuronal identity, we monitored cell proliferation and cell cycle exit following Pcdh8 IUE. We first quantified the number of mitotic progenitors among transfected cells (GFP[+]PH3[+]) and compared their positioning within the neuroepithelium of Pcdh8- and control GFP-electroporated brains. We observed a complete cell-autonomous switch from ventricular to abventricular

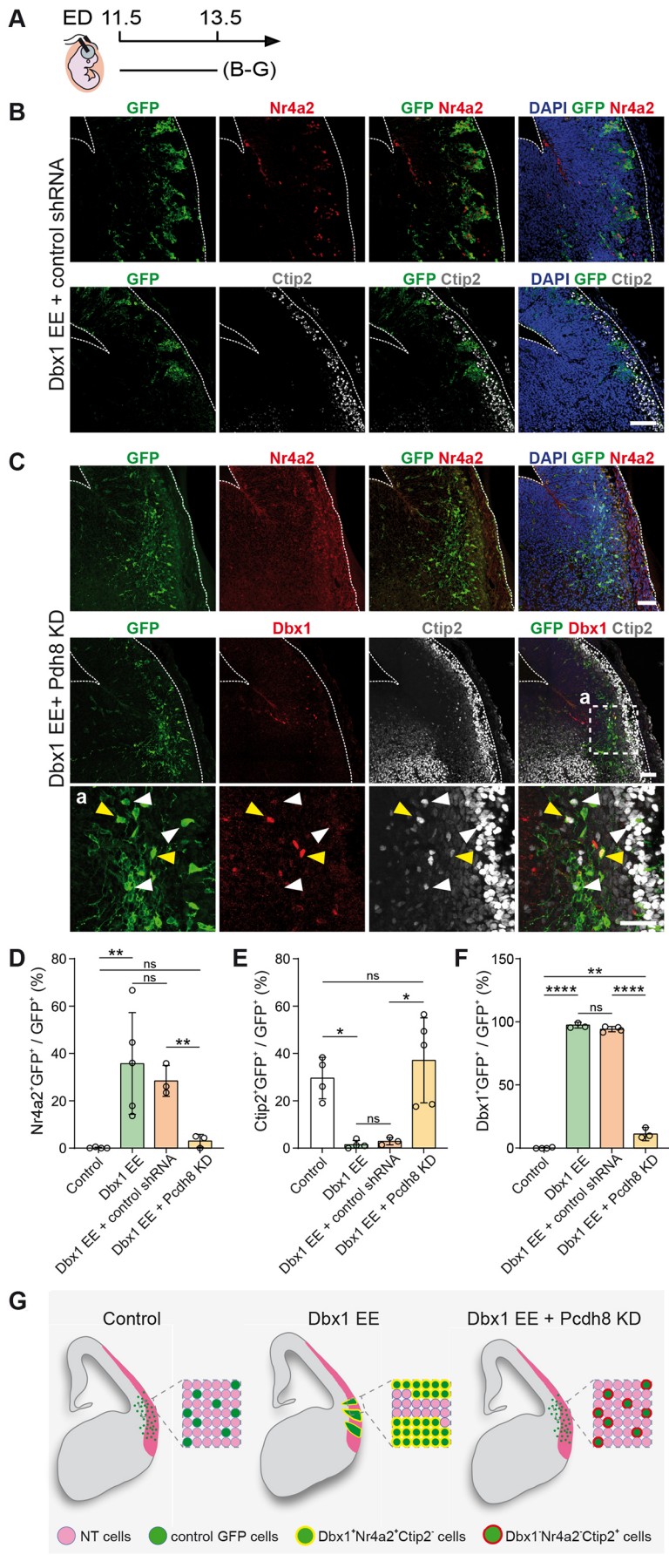

**Fig. 4. Pcdh8 is responsible for the Dbx1 EE phenotype.**
(A) Timeline of the *in utero* electroporation (IUE). ED, electroporation day. (B,C) Confocal images of GFP (green) in coronal sections of E13.5 mouse brains electroporated at E11.5 with Dbx1 EE+control shRNA or Dbx1 EE+*Pcdh8* KD (Pcdh8 shRNA), co-labelled with Nr4a2 (red) (B,C) or Ctip2 (grey) (B), or with Dbx1 (red) and Ctip2 (C). Dashed square (Ca) magnified below. White and yellow arrowheads indicate Dbx1⁻ and Dbx1⁺ cells, respectively. Dashed lines outline the pallium and ventricle. (D-F) Percentages of Nr4a2⁺GFP⁺ (D), Ctip2⁺GFP⁺ (E) and Dbx1⁺GFP⁺ (F) cells among GFP⁺ cells. Data are mean±s.e.m.; circles represent values from independent electroporated embryos (*n*=4 controls, *n*=3-5 Dbx1EE, *n*=3 Dbx1 EE+control shRNA, *n*=3-5 Dbx1 EE+*Pcdh8* KD). One-way ANOVA, post hoc Holm–Šidák: ns, not significant. (D) \*\**P*=0.0099 (control versus Dbx1 EE), \*\**P*=0.0034 (Dbx1 EE+control shRNA versus Dbx1 EE+*Pcdh8* KD); (E) \**P*<0.05 (control versus Dbx1 EE), \**P*=0.0187 (Dbx1 EE+control shRNA versus Dbx1 EE+*Pcdh8* KD); (F) \*\*\*\**P*<0.0001, \*\**P*=0.0021. (G) Scheme of *Pcdh8* KD effect on Dbx1 EE-induced aggregate formation and cell fate. Scale bars: 100 μm.

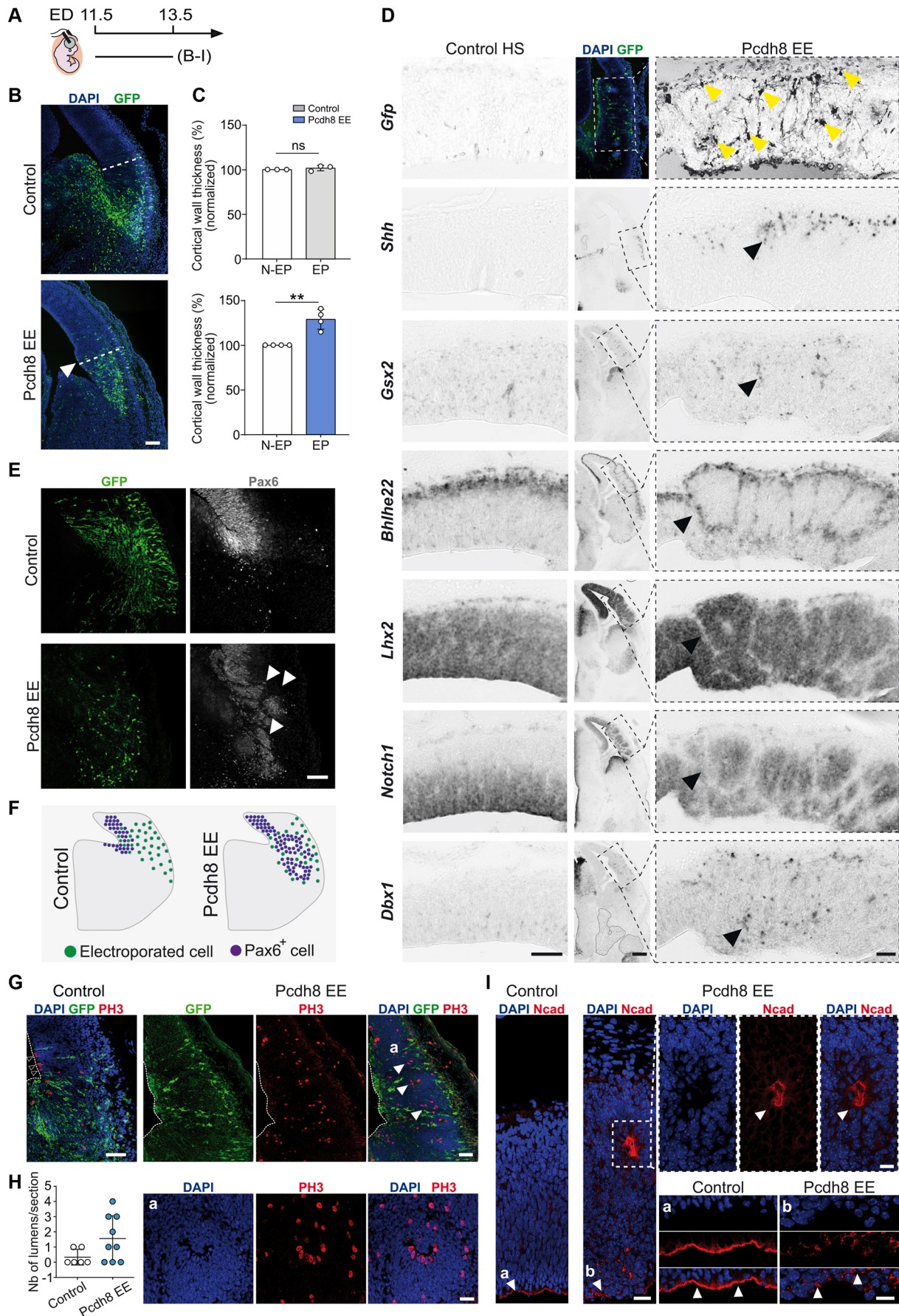

**Fig. 5.** See next page for legend.

**Fig. 5. Pcdh8 EE causes reorganisation of the developing pallium.**
(A) Timeline of the *in utero* electroporation (IUE). ED, electroporation day.
(B) Confocal images of GFP (green) in coronal sections of E13.5 mouse
brains electroporated at E11.5 with control or Pcdh8 EE, counterstained with
DAPI (blue). White arrowhead indicates the region of pallial overgrowth.
Dashed lines indicate thickness measurements. (C) Quantification of cortical
wall thickness in the electroporated (EP) region from B for control (top) and
Pcdh8 EE (bottom), normalised to the thickness of the non-electroporated
(N-EP) contralateral hemisphere. Data are mean±s.e.m. Circles represent
values from independent electroporated embryos (*n*=3 controls, *n*=4 Pcdh8
EE). Two-tailed unpaired Student's *t*-test: **$P$=0.0020, ns, not significant.
(D) Confocal image of GFP (green) in the Pcdh8 EE area (top middle),
counterstained with DAPI (blue), and bright-field ISH for *Gfp*, *Shh*, *Gsx2*,
*Bhlhe22*, *Lhx2*, *Notch1* and *Dbx1* in coronal sections of E13.5 mouse brains
electroporated at E11.5. The left panel shows the contralateral control
hemisphere (HS) from the same section as Pcdh8 EE (right panel). Dashed
rectangles from the Pcdh8 EE area magnified on the right. Yellow
arrowheads indicate cells displaying Pcdh8 EE. Black arrowheads indicate
altered mRNA expression. (E) Confocal images of GFP (green) in coronal
sections of E13.5 mouse brains electroporated at E11.5 with control or
Pcdh8 EE, co-labelled with Pax6 (grey). White arrowheads indicate the
disorganisation of Pax6$^+$ cells within the ventricular zone (VZ) and ventral
pallium (VP). (F) Scheme of Pax6$^+$ cell distribution 48 h after IUE as in
E. (G) Confocal images of GFP (green) in coronal sections of E13.5 mouse
brains electroporated at E11.5 with control or Pcdh8 EE, co-labelled with
PH3 (red) and DAPI (blue) counterstaining. Empty and white arrowheads
indicate PH3$^+$ cells aligned with or outside the VZ (dashed line),
respectively. Rosette (Ga) magnified below. (H) Quantification of rosettes in
control and Pcdh8 EE brain sections. Data are mean±s.e.m. Circles
represent values from individual sections from at least three independent
electroporations (*n*=6 controls, *n*=9 Pcdh8 EE, 1-2 sections/mouse).
(I) Confocal images of representative regions from control and Pcdh8 EE
(IUE E11.5-E13.5) labelled with Ncad (red) and DAPI (blue). Dashed square
and VZ (a,b) regions magnified at the top and bottom right, respectively.
White arrowheads indicate Ncad lining. Scale bars: 200 µm (B,D); 100 µm
(E,G,I and magnification in D); 50 µm (Ga and magnification in I).

mitotic figures upon Pcdh8 EE (Fig. S10A-E), consistent with
the observed disruption of apico-basal polarity (Fig. 5D,G,I).
Additionally, after the administration of a single pulse of EdU at
E12.5, thus 24 h post-surgery, quantification of EdU$^+$ transfected
cells revealed that Pcdh8 EE led to a ~50% reduction in S-phase
entry compared to controls (Fig. S10A,B,F) suggesting premature
cell cycle exit.

Given the massive disruption of the mitotic/postmitotic
compartmentalisation upon Pcdh8 EE (Fig. 5D-H; Fig. S8K) and
the role of the Notch pathway in maintaining the radial glia phenotype
(Blackwood, 2019) and preventing premature differentiation (Ramos
et al., 2010), we examined the expression of Notch ligands Delta1 and
Jag1 (Lasky and Wu, 2005; Stump et al., 2002) 48 h after Pcdh8 IUE
of E11.5 mouse brains (E11.5-E13.5) (Fig. 7A). We found that a non-
cell autonomous Delta1 upregulation at the centre of Pcdh8-induced
rosettes (Fig. 7B) is accompanied by a cell-autonomous decrease in
Jag1 expression (Fig. 7C). To functionally assess whether Jag1
downregulation is responsible for the observed phenotype, we
performed co-electroporation of Pcdh8 EE with a Jag1-expressing
plasmid. We showed that Jag1 EE prevented the formation of rosettes,
as well as the associated overgrowth and abnormal cell distribution,
effectively rescuing the phenotype (Fig. 7D). Together, the differential
regulation of Delta1 and Jag1 by Pcdh8 EE, along with the rescue
of the phenotype by Jag1 EE, indicate that the Notch pathway
mediates the response to Pcdh8. These findings suggest that Jag1
downregulation is responsible for altering the fate programme of
transfected progenitors (Fig. 7E) and disrupting the neuroepithelial
organisation (Fig. 7F).

Taken together, these results show that manipulating cell
adhesion through ectopic Pcdh8 expression is sufficient to affect
neuronal fate acquisition in a cell-autonomous manner via Notch
signalling alteration and premature cell cycle exit.

## DISCUSSION
### Adhesion is an important component of cell identity
TFs accomplish a key role in establishing and maintaining different
cell types during development. Indeed, changes in their expression
levels (e.g. due to ectopic expression of Pax6, Lhx2 or Dbx1) alter
initial programmes of progenitor cells (Arai et al., 2019; Pataskar
et al., 2016; Subramanian et al., 2011). Our data demonstrate that
changes in Dbx1 expression influence cellular adhesive properties
via Pcdh8, making it an essential element of cell identity that is
sensitive to Pcdh levels. Interestingly, an emerging role for cPcdhs
in the spatial and functional organisation of neurons has recently
been shown in the neocortex (Lv et al., 2022). Clustered Pcdhs are
expressed in combinatorial patterns by excitatory neurons derived
from the same progenitor cell, and these specific surface patterns
guide neuronal positioning and connectivity with other neurons (Lv
et al., 2022). However, the mechanisms regulating the expression of
distinct cPcdhs patterns remain undescribed. Here, we demonstrate
that EE of the TF Dbx1 changes the spatiotemporal competence of
the cortical neuroepithelium, consistent with its established role as a
strong cell-fate determinant across CNS regions (Arai et al., 2019;
Bouvier et al., 2010; Pierani et al., 2001; Sokolowski et al., 2016).
Specifically, we found that Dbx1 EE imparts different cell fates in a
time- and region-dependent manner. Temporally, it promoted the
generation of Nr4a2$^+$Ctip2$^-$ cells (not subplate neurons) at E11.5
and Nr4a2$^-$Ctip2$^+$ at E12.5 in the LP. Regionally, at E11.5, it gives
rise to Cajal-Retzius and subplate neurons in the DP (Arai et al.,
2019), while in the LP/VP it induced Nr4a2$^+$Ctip2$^-$ cells, and not
the dorsal pallial marker Bhlhe22, likely characteristic of the CLA
(Mantas et al., 2024; Yan et al., 2025 preprint). These distinct
outcomes correlated with the induction of distinct Pcdhs, namely
Pcdh8 at E11.5 and Pcdh9/Pcdh19 at E12.5, suggesting that
progenitor cell competence varies over time, producing daughter
cells with different Pcdh expression combinations on their surface.
Consistently, our expression analyses show that δ-Pcdhs (Pcdh8,
Pcdh9, Pcdh19) display sharply distinct and dynamic spatial
patterns between E11.5 and E13.5. Importantly, δ-Pcdhs mediate
strictly homophilic adhesion and avoid heterotypic interactions
(Bisogni et al., 2018). Even subtle mismatches in δ-Pcdh expression
can prevent co-aggregation, implying that adhesive affinity and
surface composition determine segregation behaviour. This
principle likely explains why Dbx1 EE in the LP leads transfected
cells into Pcdh8-expressing aggregates with CLA-like neuron
profile (Nr4a2$^+$Ctip2$^-$Bhlhe22$^-$Reln$^-$Calb2$^-$), whereas in the DP it
did not induce aggregation but instead correlated with Cajal-Retzius
and subplate-like neuron fates (Arai et al., 2019). We propose that
high Pcdh8 expression in Dbx1 EE-expressing cells, combined with
low Pcdh19 levels preferentially in LP/VP progenitors, promoted
strong homophilic interactions and aggregate formation *in vivo*.
Conversely, higher co-expression of other δ-Pcdhs (i.e. Pcdh19) in
the DP likely reduces aggregation due to heterotypic incompatibility.
Together, we believe that the strong cell-segregative character of
Pcdh8, combined with dynamic developmental changes in adhesion
molecule combinatorial codes, regional (medial-dorsal-lateral axis)
and progenitor-specific differences along the DV axis (i.e. VP and
septum versus subpallium), modulate adhesion strength (Bisogni
et al., 2018) and possibly neuronal migration, thereby contributing to
the segregation of distinct neuronal populations. Moreover, variations

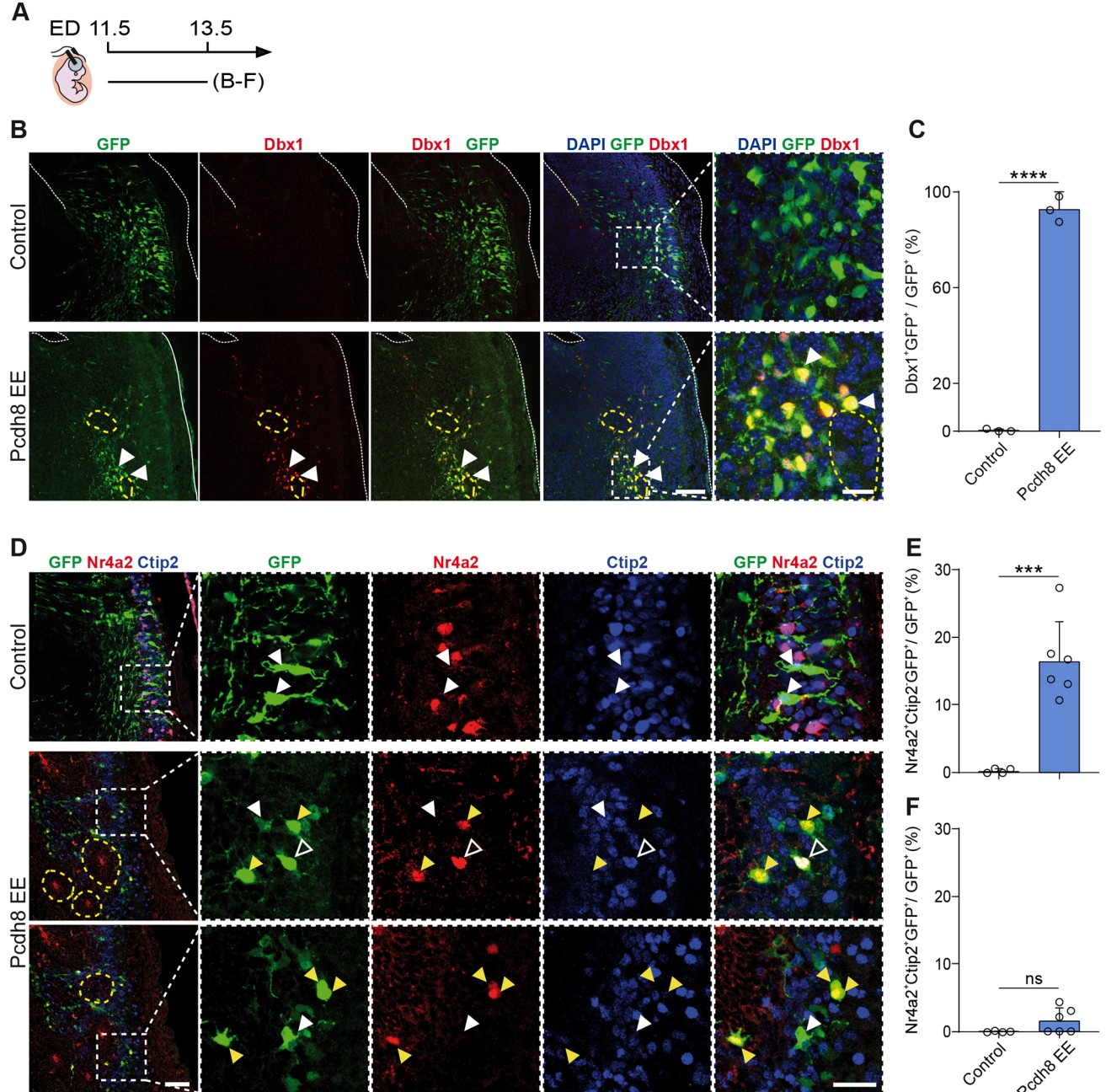

**Fig. 6. Pcdh8 EE induces Dbx1 and Nr4a2+ CLA-like neuron fate.** (A) Timeline of the *in utero* electroporation (IUE). ED, electroporation day. (B) Confocal images of GFP (green) in coronal sections of E13.5 mouse brains electroporated at E11.5 with control or Pcdh8 EE, co-labelled with Dbx1 (red) and DAPI (blue) counterstaining. White arrowheads indicate Dbx1+GFP+ cells. Dashed squares highlight GFP+ regions magnified on the right. Dashed lines outline the pallium and ventricle. (C) Percentage of Dbx1+GFP+ cells among total GFP+ cells. Circles represent values from independent electroporated embryos (*n*=3 each condition). (D) Confocal images of GFP (green) in coronal sections of E13.5 brains electroporated at E11.5 with control or Pcdh8 EE, co-labelled with Nr4a2 (red) and Ctip2 (blue). Dashed boxes are magnified on the right. Yellow, empty and filled white arrowheads indicate Nr4a2+Ctip2−GFP+, Nr4a2+Ctip2+GFP+ and Ctip2+Nr4a2−GFP+ cells, respectively. (E,F) Percentages of Nr4a2+Ctip2−GFP+ (E) and Nr4a2+Ctip2+GFP+ (F) cells among GFP+ cells. Circles represent individual section values (*n*=3 electroporations/condition, 1-2 sections/mouse). Dashed yellow lines outline Pcdh8 EE-induced rosette-like structures. Data are mean±s.e.m. Two-tailed unpaired Student's *t*-test: ****P<0.0001 (C), ***P=0.0006 (E), ns, not significant (F). Scale bars: 200 µm (B); 100 µm (D); 50 µm (magnification in B and D).

in adhesion properties can play a crucial role in organising properly functioning brain areas. Pcdh8 EE resulted in the reorganisation of expression of multiple TFs crucial for correct DV patterning (e.g. *Shh*, *Dbx1*, *Lhx2*), suggesting reciprocal regulation between adhesion molecules and transcriptional networks. Notably, during early stages of spinal cord development, specific cell adhesion molecules

have been shown to follow the expression of TFs in selected progenitor domains (Tsai et al., 2020), ensuring robust tissue patterning. Altogether, our findings suggest that Dbx1 expression is both temporally and spatially restricted and, together with Pcdh8, can influence TF spatial organisation, thereby contributing to proper cortical development.

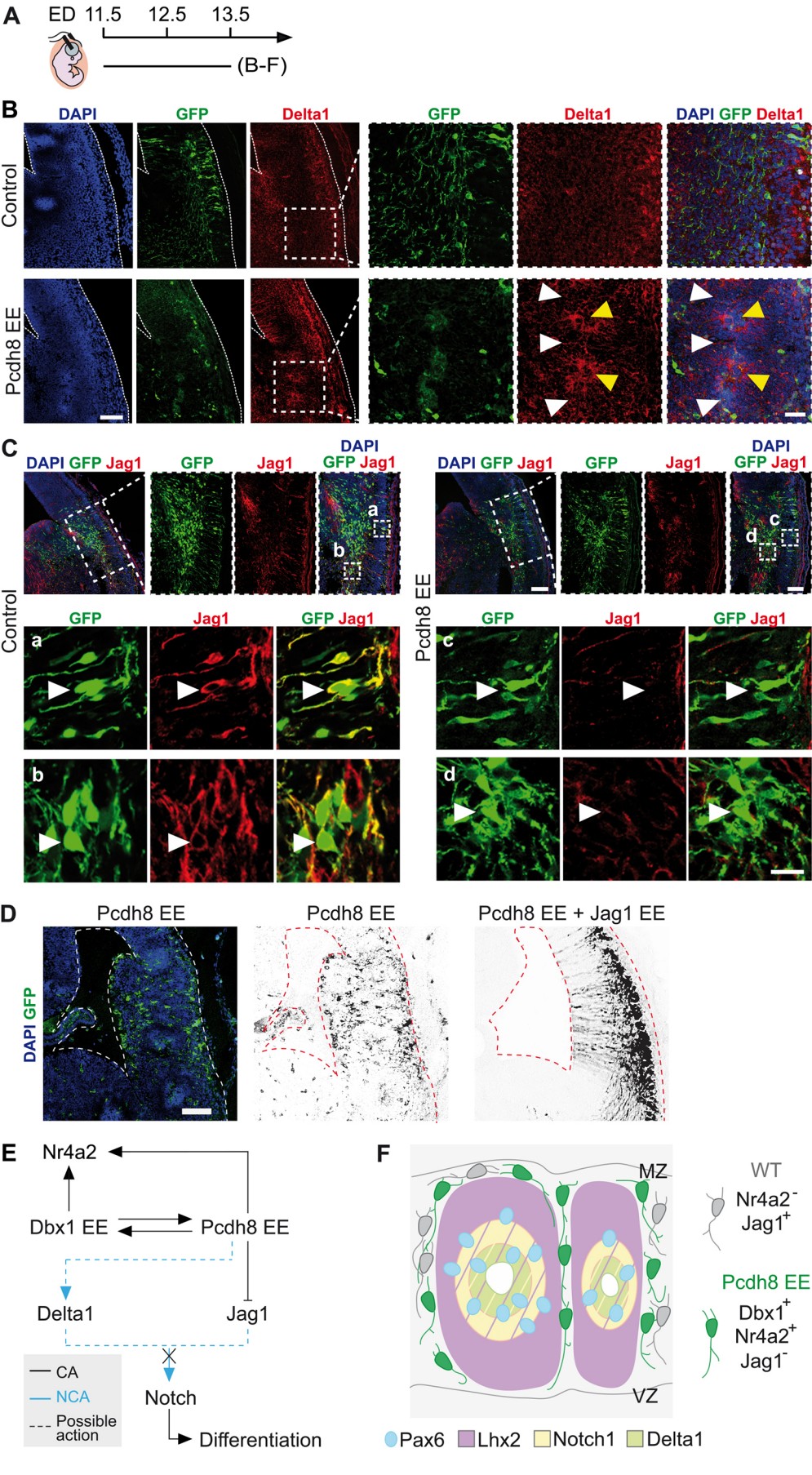

**Fig. 7. Pcdh8 EE phenotype is mediated by the Notch1 pathway.** (A) Timeline of the *in utero* electroporation (IUE). ED, electroporation day. (B) Confocal images of GFP (green) in coronal sections of E13.5 mouse brains electroporated at E11.5 with control or Pcdh8 EE, co-labelled with Delta1 (red) and DAPI (blue) counterstaining. Dashed squares are magnified on the right. Yellow and white arrowheads indicate Delta1 accumulation or surrounding GFP+ cells with low Delta 1 signal, respectively. Dashed lines outline the pallium and ventricle. (C) Confocal images of GFP (green) in coronal sections of E13.5 mouse brains electroporated at E11.5 with control or Pcdh8 EE, co-labelled with Jag1 (red) and DAPI (blue) counterstaining. Dashed rectangles magnified on the right. Insets in control (Ca,Cb) and Pcdh8 EE (Cc,Cd) magnified in the bottom panels. White arrowheads indicate representative cells of each condition. (D) Confocal images of coronal sections of E13.5 mouse brain cortices electroporated at E11.5 with Pcdh8 EE alone or co-electroporated with Jag1 EE (1:1 ratio). GFP (green) and DAPI (blue) staining and grayscale GFP images are shown. Dashed lines delineate the pallium and ventricle. (E) Scheme of proposed interactions between Dbx1, Pcdh8, Notch signalling and cell fate markers. CA, cell autonomous; NCA, non-cell autonomous. (F) Cartoon depicting rosette-like structure composition formed upon Pcdh8 EE. Scale bars: 200 μm (B,C); 100 μm (D and magnification in C); 50 μm (magnification in B); 25 μm (Ca-Cd).

## Crosstalk between Pcdh8 and TFs determines CLA-like fate

Dbx1 was previously shown to function as a potent determinant of cell identity in the mouse spinal cord (Pierani et al., 2001). During early development, Dbx1-expressing domains are present at the borders of the pallium and subpallium, notably the VP/PSB and septum (Bielle et al., 2005; Griveau et al., 2010). Interestingly, we detected co-expression of Dbx1 and Pcdh8 in VP and septal progenitors, and Dbx1-derived cells in the postmitotic compartment of the septum and VP in a ventrally migrating stream toward the prospective CLA. We showed changes in *Pcdh8* expression domains in Dbx1$^{lacZ/lacZ}$ KO mice, notably a reduction in the *Pcdh8*$^+$ postmitotic compartment, suggesting that Dbx1 regulates Pcdh8 expression in a region-specific manner, restricting its expression in septal progenitors while promoting Pcdh8-expressing neuronal differentiation in the VP. This pattern correlates with the emergence of the lateral postmitotic Pcdh8$^+$ (Nr4a2$^+$) population; however, this population appears to be partially lost in the absence of Dbx1, likely resulting from impaired cell-cycle exit (García-Moreno et al., 2018). Moreover, genetic lineage tracing in the developing mouse cortex demonstrated that Dbx1 progenitors also give rise to Nr4a2$^+$ and Pcdh8$^+$ neurons located in territories corresponding to the future CLA. These data argue in favour of a potential direct, though not exclusive, interaction between Dbx1 and Pcdh8 in cell identity determination. At the molecular level, *Pcdh8* KD in Dbx1 EE cells prevented Dbx1 expression and consequently reversed CLA-like cell identity, whereas Pcdh8 EE induced both Dbx1 expression and CLA-like fate. While Dbx1 may control Pcdh8 expression at the transcriptional level, the use of the same pCAGGS promoter in both EE and KD experiments, together with GFP expression confirming successful transcription and translation, suggests that Pcdh8, in turn, modulates Dbx1 post-transcriptionally. Overall, these results clearly show that Dbx1 and Pcdh8 interact in a reciprocal manner within the VP/LP to specify CLA-like neuron identity. Complementing Arai et al. (2019), our data confirm that in mice, Dbx1$^+$ progenitors do not generate subplate neurons, unlike in primates, and identify the bidirectional molecular interaction between Dbx1 and Pcdh8 that mediates Nr4a2$^+$ CLA-like neuronal fate. This provides a previously unrecognised and important role for adhesion molecules in regulating TF expression and cell fate, providing key insights into cortical development and functional organisation.

## Pcdh8 regulates cortical development from cell to tissue organisation levels

Disturbance of TF expression such as Lhx2 and/or Pax6 results in re-patterning of TF expression domains (Mangale et al., 2008; Roy et al., 2014; Godbole et al., 2017) and tissue overgrowth (Mangale et al., 2008). We found that Dbx1 EE influenced cell fate by inducing Pcdh8 expression without major DV patterning defects. However, shifting Pcdh8 expression, typically restricted to the septum and/or ventrolateral domains, into more dorsal pallial regions disrupted the expression of TFs Lhx2 and Pax6 in progenitor cells. This was accompanied by AP delamination through loss of Ncad$^+$ adherens junctions in the neuroepithelial cells lining the ventricle, and formation of rosette-like structures containing new proliferative centres enriched with Ncad. Moreover, we observed ectopic expression of the subpallial progenitor TF marker *Gsx2* in the DP, while the pallial postmitotic *Bhlhe22* localised to the periphery of the rosettes, both in a cell-autonomous or non-cell autonomous manner. These observations are more profound but still consistent with studies on disruptions of the Gsx2-Pax6-Dbx1 border at the PSB (Cocas et al., 2011) or the hem-antihem balance due to Lhx2 dysregulation, which dramatically impair DV organisation and, consequently, deform the developing pallium

(Godbole et al., 2017). Since Pcdh expression localised to apical membranes (Lobas et al., 2012) and Ncad is crucial for neuroepithelial integrity and cortical organisation (Kadowaki et al., 2007), our work adds Pcdh8 as a new DV organiser. The observed phenotypic differences between Dbx1 EE and Pcdh8 EE likely derive from the timing of Pcdh8 expression, whether directly driven by the pCAGGS promoter/enhancer or induced later by Dbx1 EE (itself being first expressed under pCAGGS control). As Dbx1 promotes cell cycle exit (García-Moreno et al., 2018), pCAGGS-driven Pcdh8 will induce expression directly in APs, whereas pCAGGS-driven Dbx1 expression in APs will trigger rapid cell cycle exit, thus likely inducing Pcdh8 expression in young neurons (or BPs). Mechanistically, Pcdh8 may regulate TF expression, notably Dbx1, through post-transcriptional mechanisms mediated by its intracellular domain (ICD). The ICD links cadherin-mediated adhesion to Wnt signalling, crucial for cell fate, migration and tissue patterning (Pancho et al., 2020), by activating TAO2β and p38-MAP kinases, which regulate nuclear speckle function, RNA processing and post-transcriptional gene expression (Gao et al., 2003; Mahtani et al., 2001; Yasuda et al., 2007). Pcdh8 also modulates both non-canonical and canonical Wnt/β-catenin and AKT/GSK3β/β-catenin pathways (Unterseher et al., 2004; Zong et al., 2017). GSK3, a central regulator of neuronal development (Gizak et al., 2020), controls cadherin mRNA stability through β-catenin-dependent and -independent mechanisms (Farina et al., 2009; Gizak et al., 2020; Unterseher et al., 2004; Zong et al., 2017). Moreover, Pcdh8 activates Wnt/planar cell polarity (PCP) signalling, coordinating cell polarity and morphogenesis (Kraft et al., 2012; Medina et al., 2004). Its ICD further interacts with Sprouty, NLK1 and CK2β signalling factors to fine-tune Wnt signalling and Pcdh8 stability, while Pcdh8 expression itself is regulated by Wnt/PCP activity, forming a feedback loop (Schambony and Wedlich, 2007). Collectively, these results demonstrate that Pcdh8, which is normally strongly expressed in septal and VP/PSB progenitors, when ectopically expressed at or dorsal to the PSB can profoundly affect DV organisation and TF activity in both cell-autonomous and non-cell autonomous manners.

## Notch signalling as a downstream partner of Pcdh8

The Notch signalling pathway tightly controls both the temporal and spatial patterning of neuronal populations (Wang et al., 2015; Ware et al., 2016). Notch1 inactivation has been associated to accelerated neuronal differentiation in the mouse spinal cord and developing brain (Kong et al., 2015; Tokunaga et al., 2004; Yang et al., 2006). Additionally, it controls the number of neural progenitor cells that exit the cell cycle and undergo differentiation (Moore and Alexandre, 2020). We showed that Pcdh8 influences cell cycle exit and postmitotic fate acquisition, and that it represses Notch1 signalling ligands Jag1 and Dll1 in cell-autonomous and non-cell autonomous manners, respectively, thus suggesting a potential impact on differentiation. A rescue experiment combining Pcdh8 EE with Jag1 EE further demonstrated that restoring Jag1 in Pcdh8 EE restored normal pallial formation, providing strong evidence that Jag1 downregulation underlies the Pcdh8 EE-induced phenotype. Interestingly, Dbx1 was also shown to influence Jag1 expression in the spinal cord (Skaggs et al., 2011) and promotes cell cycle exit (García-Moreno et al., 2018). Together, these findings position the Notch pathway as a relevant candidate for the cooperative function between Pcdh8 and Dbx1, which pushes progenitors out of the cell cycle toward premature differentiation.

In summary, our study establishes adhesion molecules, notably Pcdh8, as newly discovered players in pallium morphogenesis and cell fate determination. Above all, we demonstrate that Pcdh8,

through a bidirectional regulatory mechanism with the TF Dbx1, governs cell cycle dynamics and fate acquisition, ultimately shaping cortical organisation. Moreover, by identifying Notch signalling as a downstream effector of Pcdh8, our findings shed light on the regulatory mechanism coordinating adhesion, transcriptional control and neurogenesis, providing new insights into the molecular orchestration of neural development.

## Study limitations

The main limitation is the lack of specific embryonic CLA markers to formally demonstrate that the identity of Nr4a2[+]-induced neurons fully recapitulate mature CLA fate. Most known CLA markers are defined in adults (Fodoulian et al., 2025; Hara et al., 2025; Kaur et al., 2025), and developmental trajectories linking embryonic and mature populations remain unresolved. Nr4a2 thus remains the most representative early marker for CLA-like neurons. Dbx1-derived CLA neurons in the mouse represent a small subset of the Nr4a2[+] population (Puelles et al., 2016), requiring other early markers to investigate changes in cell identity in the Dbx1 KO. Furthermore, studies using ectopic gene expression or KD strategies bear intrinsic drawbacks, such as non-physiological protein levels or potential off-target effects. Despite these limitations, our findings demonstrate a model in which TFs and cell adhesion molecules interact bidirectionally to define cell identity during cortical development.

## MATERIALS AND METHODS

### Animals

The following mouse lines were used and maintained on a *C57BL/6J* background: *Dbx1[lacZ]* (Pierani et al., 2001), *Rosa26[YFP]* (Srinivas et al., 2001) and *Rosa26[tdTomato]* (Madisen et al., 2012). All animals were handled in strict accordance with good animal practice as defined by the national animal welfare bodies, and all mouse work was approved by the Veterinary Services of Paris (authorisation number 75-1454) and by the Animal Experimentation Ethical Committee Paris-Descartes(CEEA-34) (reference 2018012612027541).

### Plasmid DNA cloning

For the study we used a pCAGGS-HA-mouseDbx1-ires-NLS-EGFP vector previously published in Arai et al. (2019). The Pcdh8 overexpression sequence was subcloned into the pCAGGS vector backbone using In-Fusion® HD Cloning (Takara Bio) accordingly to the manufacturer's instructions.

Pcdh8 shRNAs and scrambled shRNA sequences were designed and subcloned into psiSTRIKECAG-ires-GFP vectors (a gift from Dr Billuart, Université Paris Cité, IPNP, INSERM U1266, and Institut Imagine, Paris, France) using In-Fusion® HD Cloning (Takara Bio) according to the manufacturer's instructions. The most efficient shRNA, reducing Pcdh8 expression by 51% in HEK293T cells 48 h after co-transfection with the *Pcdh8* expression vector, was identified by western blot analysis.

### In utero electroporation

The procedure was carried out following previously published protocols (Arai et al., 2019; Szczurkowska et al., 2016). Briefly, to ensure the welfare of the animals used, 30 min before the surgery 50 µg/kg of Buprenorphine followed by 5 mg/kg Ketoprofen post-surgery subcutaneous administration were performed. WT *C57BL/6J* pregnant mice at E11.5 or E12.5 were subjected to incision under anaesthesia with Isoflurane (AXIENCE SAS), and the uterine horns were exposed onto 1× phosphate-buffered saline (PBS)-moistened cotton gauze. Embryos were visualised using appropriate flexible light sources through the uterus. Plasmid DNAs mixed with a filtered Fast Green dye were injected into the lateral ventricle through a glass capillary. A pair of 3 mm electrodes was applied to the embryos through the yolk sac, and a series of square-wave current pulses (25 V, 50 ms) was delivered six times at 950 ms intervals using a pulse generator (NEPA21, NEPAGENE). Uterine horns were repositioned into the abdominal cavity, and the abdominal wall and skin were sutured. The concentration of plasmid

DNAs used was between 1 and 3 µg/µl. The animals were checked in the post-surgery care room (heating pad, wet food and, if necessary, additional dose of painkiller was administrated).

### Immunofluorescence

Mice were killed by cervical dislocation and embryos were collected and fixed at 4°C for 2 h (E11.5-E12.5), 2.5 h (E13.5), 3 h (E14.5) or 4 h (E18.5) in 4% paraformaldehyde (PFA)/PBS, washed in PBS 3×1 h at 4°C, cryoprotected in 30% sucrose/PBS overnight at 4°C, and embedded in Tissue-Tek O.C.T compound (Sakura Finetek). Coronal cryosections of mouse embryos (20 µm, E11.5-E14.5; 25 µm, E18.5) were blocked with 0.2% Triton X-100 in 10% horse serum/1× PBS for 30 min. For Pcdh8 immunostaining, before blocking, sections were incubated for 10 min at 95°C in antigen retrieval solution (10 mM sodium citrate, pH 6.0) and cooled down for 20 min at room temperature (RT). Sections were subsequently incubated overnight at 4°C with primary antibodies, followed by incubation with fluorescently labelled secondary antibodies and DAPI in 0.2% Triton X-100, 1% horse serum, 1× PBS buffer for 45 min at RT. Sections were mounted with Vectashield mounting medium. Primary antibodies used were: Bhlhe22, 1:200; Calb, 1:1000; Ctip2, 1:2000; GFP, 1:2000; Dbx1, 1:10,000; Delta1, 1:100; Jag1, 1:500; Ncad, 1:400; Nurr1/Nr4a2, 1:200; Pax6, 1:1000; Pcdh8, 1:700; Pcdh19, 1:500; PH3, 1:500; Reln, 1:1000; Tle4, 1:500. Secondary antibodies used were: Alexa 488, 1:700; Cy3, 1:700; Cy5, 1:500 (see Table S1 for full details of antibodies and reagents).

### cDNA synthesis and qPCR

RNA samples from electroporated tissue were retro-transcribed to cDNA using the RevertAid First Strand cDNA Synthesis Kit, according to the supplier's instructions. For qPCR analysis of gene expression in each electroporated region, we used 500 ng of RNA. The genomic DNA was removed by incubation for 30 min in 37°C with DNase I, RNase-free. All qPCR reactions were performed on an Eppendorf machine in 20 µl reaction volume of Go Tag qPCR Master Mix (1× SYBR®Green PCR Master Mix and 200 nM of each primer). The following thermal programme was applied: denaturation for 2 min at 95°C followed by 40 amplification cycles of 15 s denaturing step (95°C) and 1 min annealing–extension step (60°C). Afterwards, to verify that results are coming from the single transcript, the automatic programme for melting curve analysis was performed using standard machine settings. Expression data were normalised to the geometric mean of housekeeping gene *Gapdh*, which showed stable expression across all samples. The correlation plots in Fig. S6A depict gene expression levels relative to *Dbx1* expression in each electroporated sample, whereas the analyses in Fig. 3C-F and Fig. S6D,E represent mean gene expression values, normalised to *Gapdh*, across all control and Dbx1 EE samples.

### *In situ* and fluorescent *in situ* hybridisation

ISH was performed as previously described (Griveau et al., 2010). For each gene of interest, a DNA fragment (typically 500-800 bp) was amplified by PCRs from an embryonic brain cDNA library using Phusion polymerase (Thermo Fisher Scientific) and primers indicated in Table S1. The promoter sequence of the T7 RNA polymerase (GGTAATACGACTCACTATA-GGG) was added 5′ of the reverse primers. Antisense RNA probes were labelled using a DIG-RNA labelling mix (Roche). Alternatively, for mouse *Gsx2*, *Lhx2* (gift from Dr S. Retaux, CNRS and the Université Paris-Saclay, Gif-sur-Yvette/Saclay, France), *Notch1* (gift from Prof. U. Lendahl, Karolinska Institutet, Stockholm, Sweden) and *Shh* (gift from Dr S. Garel, Collège de France, IBENS and CIRB, Paris, France), a plasmid containing part of the cDNA was linearised by restriction enzyme digestion and subsequently submitted to reverse transcription. Sections were mounted with Mowiol 4-88. For FISH, antisense RNA probes were labelled using either the DIG-RNA labelling mix (Roche) or the Fluorescein labelling mix (Roche). Fluorescent detection was performed using Anti-FITC-POD (Roche) or Anti-DIG-POD (Roche), followed by signal amplification with Alexa Fluor™ 488 Tyramide SuperBoost™ (Invitrogen) or Cyanine 3 Tyramide (Biotechne), respectively (Table S1). Co-localisation of *Pcdh8* (red signal) and *Dbx1* (green signal) was identified in cells outlined by DAPI-stained nuclei.

## Image acquisition and quantifications

*In situ* hybridisation images were obtained using a Hamamatsu Nanozoomer 2.0HT slide scanner with a 20× objective. The scans were verified using NDP Viewers and imported into Adobe Photoshop for figure adjustments.

All fluorescence images and tile scan images were acquired using a Leica TCS SP8 SMD confocal microscope equipped with a 40× objective, analysed with ImageJ and imported into Adobe Photoshop for figure adjustments. Using ImageJ, cluster quantification was performed across the MP, DP and LP regions, each defined using consistent region of interest (ROI) masks. GFP$^+$ cell density per ROI was calculated by dividing the total number of GFP$^+$ cells (identified using DAPI nuclear counterstaining) by the ROI area. The number of clusters (defined as groups of more than three closely associated cells) was normalised to GFP$^+$ cell density within each ROI. For cell identity, the proportion of the marker-positive cells among the GFP$^+$ was quantified using ImageJ. Data shown in Fig. 1E,F, Fig. S1C, Fig. 4D,E,F, Fig. S2F and Fig. S9D for control and/or Dbx1 EE originate from the same set of experiments and are included in multiple figures to enable cross-comparisons while minimising animal use.

## EdU pulse labelling and staining

EdU injection was carried out by intraperitoneal injection of 100 μl of 1 mg/ml EdU (Invitrogen) in PBS into pregnant females at E12.5 after electroporation at E11.5. IF and EdU staining was performed using Click iT EdU Alexa Fluor 647 Imaging Kit (Invitrogen).

## Cell culture, transfection, sample preparation and adhesion assay

Human embryonic kidney (HEK) 293T cells were grown in Dulbecco's modified Eagle's medium (DMEM; Gibco) containing 10% foetal bovine serum (FBS) and 1% penicillin-streptomycin (P/S; Gibco) and maintained at 37°C in a 5% $CO_2$ atmosphere. Cells were seeded onto six-multiwell plates at a density of 62,500 cells/cm$^2$. Then, 24 h later, transfection with 2 μg of pCAGGS constructs containing control GFP or Pcdh8, and with 4 μg of Pcdh8 scrambled shRNA or Pcdh8 shRNA, was performed for 4 h using Lipofectamine 2000 (Invitrogen) according to the manufacturer's instructions. Non-transfected cells (treated with transfection reagent only) were also used as control. Cells were maintained in serum-free Opti-MEM (Gibco) supplemented with 1% P/S after transfection and cultured for an additional 40 h. Cell lysates were collected using RIPA buffer (50 mM Tris-HCl pH 7.5, 150 mM NaCl, 1% NP-40, 0.1% SDS, 0.5% sodium deoxycholate) supplemented with 2 mM EDTA and protease inhibitors (cOmplete™ EDTA-free tablets, Roche). Samples were then left in a rotator for 30 min at 4°C to better dissolve the proteins and centrifuged at maximum speed for 10 min at 4°C in order to pellet out crude undissolved fractions. The supernatant was stored at −20°C until use. Protein quantification was determined using the bicinchoninic acid (BCA) protein assay reagent kit (Pierce™, Thermo Fisher Scientific) using bovine serum albumin as standard.

L929 cells were plated and maintained in DMEM supplemented with 1% L-glutamine and 10% FBS. Cultures were incubated at 37°C in a 5% $CO_2$ humidified incubator.

Adult subependymal zone (SEZ) neural stem cell (NSC) cultures were obtained from 2- to 3-month-old mice, following the protocol described by Belenguer et al. (2016). NSCs were cultured in DMEM/F-12 (Gibco) medium supplemented with 20 ng/ml epidermal growth factor (EGF, Gibco), 10 ng/ml basic fibroblast growth factor (bFGF, Sigma-Aldrich), and B27 supplement (Thermo Fisher Scientific). Cultures were maintained for 7-10 days at 37°C in a 5% $CO_2$ humidified incubator.

For adhesion experiments, L929 cells overexpressing N-cadherin were seeded onto glass coverslips and allowed to reach confluence over 24-48 h. Passage 2 NSCs were freshly dissociated from the SEZ of three adult WT mice (2-3 months old) and transfected with either a Control or Dbx1 EE vector using the NSC Nucleofection Kit (Lonza) and Amaxa Nucleofector II (Amaxa), following the manufacturer's instructions. After 24 h, transfected cells were analysed via FACS (BD LSRFortessa™ Cell Analyzer) to determine the percentage of GFP$^+$ cells. Subsequently, NSCs were seeded at a

density of 1.25×10$^4$ cells/cm$^2$ onto the pre-formed L929 monolayers. After 45 min of incubation, cultures were thoroughly washed to remove unattached cells, followed by fixation and analysis.

## Western blot analysis

Protein samples were added to 1/4 volume of 4× LDS sample buffer (141 mM Tris, 106 mM Tris-HCl, 2% LDS, 10% glycerol, 0.51 mM EDTA, 0.22 mM SERVA Blue G, 0.175 mM Phenol Red, pH 8.5), to 1/10 of sample reducing agent 10× (50 mM DTT) and boiled at 95°C for 5 min. Protein samples (10 μg of lysates) were separated by SDS-PAGE on 3-8% tris-acetate gels (NuPAGE, Invitrogen) under reducing conditions at 150 V at 4°C and electro-transferred to 0.45 μm nitrocellulose membranes for 1.5-2 h at 0.5 A at 4°C. After 1 h blocking at RT with 5% (w/v) milk in Tris-buffered saline (50 mM Tris, 150 mM NaCl, pH 7.6) containing 0.1% Tween-20 (TBS-T), the membranes were incubated overnight at 4°C with the following primary antibodies: Pcdh8 (1:1000) and αTubulin (1:10,000). After three washes of 10 min with TBS-T, the membranes were incubated with the appropriate HRP-conjugated secondary antibodies (1:20,000) diluted in TBS-T with 5% (w/v) milk for 1 h at RT. After three 10 min washes with TBS-T, blots were developed with SuperSignal West Pico Chemiluminescent Substrate (Thermo Fisher Scientific) and visualised on the ChemiDoc apparatus (Bio-Rad). Densitometry analysis was determined using Image Lab™ software (Bio-Rad). The densities of protein bands were quantified with background subtraction. The bands were normalised to αTubulin loading control. The molecular weights were determined by using an appropriate pre-stained protein standard (HiMark 31-460 kDa, Invitrogen).

## Microarray and scRNAseq analysis

*Dbx1$^{Cre}$;Rosa26$^{YFP}$* cell sorting and microarray analysis at E12.5 were performed as previously described in Griveau et al. (2010). scRNAseq was performed as previously described for the VP (Moreau et al., 2021). The entire telencephalic vesicle dataset was generated using three WT embryos aged E11.5-E12 from two distinct litters. For the septum analysis, explants encompassing the septum were dissected and pooled from eight E11.5 *PGK$^{Cre}$;Rosa26$^{YFP}$* embryos originating from two distinct litters and four E12.5 *Dbx1$^{Cre}$;Rosa26$^{tdTomato}$* embryos from two litters. The dissected tissue was dissociated using the Neural Tissue Dissociation Kit (P) (Miltenyi Biotec) and a gentleMACS Octo Dissociator following the manufacturer's instructions. Cell clumps and debris were removed via filtration through 30 μm cell strainers (Miltenyi Biotec) followed by two consecutive rounds of centrifugation for 3 min at 200 ***g***. Pipetting through Gel Saver tips (QSP) and a final filtration using a 10 μm cell strainer (pluriSelect) allowed us to obtain a single-cell suspension. Approximatively 10,000 cells were used as input on the 10x Genomics Chromium controller. A Single Cell 3′ Kit v2 library was produced and sequenced on a NextSeq500 sequencer at a total depth of 476 million reads. Raw sequencing reads were processed to count matrix with Cell Ranger version 2.1.1 using default parameters and the mm10 mouse genome reference to which the sequences of *YFP* and *tdTomato* were added. Bioinformatic analyses were performed with R version 4.1.1. Cells were filtered based on the percentage of mitochondrial reads, removing those outside three median absolute deviation (MAD) from the median. The count matrix was library size normalised using Seurat V4.1.0 (Hao et al., 2021). Doublet removal was achieved using Scrublet (Wolock et al., 2019). In some instances, the SPRING tool (Weinreb et al., 2018) was used for dimensionality reduction. Gene expression curves in Fig. 2D-E represent normalised gene expression, calculated using the Seurat function 'NormaliseData()' with the parameters normalisation.method='LogNormalise' and scale.factor=10,000. The curves were plotted using the ggplot2 function 'geom_smooth()' with default parameters.

## Statistical analysis

All data are expressed as mean±s.e.m. *P*-values <0.05 were considered significant and set as follows: \*$P<0.05$, \*\*$P<0.01$, \*\*\*$P<0.001$, \*\*\*\*$P<0.0001$. According to the data structure, two-group comparisons were performed using two-tailed unpaired Student's *t*-test or Holm–Šidák as post-hoc test following one-way ANOVA. Statistics and plotting were performed using GraphPad Prism 7.0.

## Acknowledgements

We thank E. Panafieu, Animalliance and the Imagine Institute animal house staff for mouse care and technical help. We also appreciate advice on the cloning process and donation of pStrike plasmid by P. Billuart. Furthermore, we thank Stéphane Nedelec, Sophie Thomas, Jose Manuel Morante Redolat and Isabel Martinez Garay for their help in the experiments not included in this manuscript. Finally, we thank Isabel Fariñas for providing the laboratory space to perform some of the experiments and Cristina Gil-Sanz for the Tle4 antibody.

## Competing interests

The authors declare no competing or financial interests.

## Author contributions

Conceptualization: A.P., A.W.C., S.F.; Data curation: A.W.C., J.G.-J.; Formal analysis: A.W.C., S.F., E.D., J.G.-J.; Funding acquisition: A.P.; Investigation: A.P., A.W.C., S.F., E.D., J.G.-J., M.X.M., Y.S., P.G.-B., S.C.-P., J.G.-M., D.M., U.B., F.C.; Methodology: A.P., A.W.C.; Project administration: A.P.; Resources: A.P.; Supervision: A.P.; Validation: A.P., A.W.C., S.F.; Visualization: A.P., A.W.C., S.F.; Writing – original draft: A.P., A.W.C., S.F., F.C.; Writing – review & editing: A.P., A.W.C., S.F., F.C.

## Funding

This work was supported by Fondation pour la Recherche Médicale (SPF20170938863  to A.W.C.), Fondation Fyssen (CH9147 to A.W.C.), Generalitat Valenciana (CDEIGENT/2021/005 to A.W.C.), by grants from the Agence Nationale de la Recherche (ANR-2011-BSV4-023-01, ANR-15-CE16-0003-01 and ANR-20-CE16-0001-01), Fondation pour la Recherche Médicale (Equipe FRM DEQ20130326521 and EQU201903007836) to A.P, and state funding from the Agence Nationale de la Recherche under the 'Investissements d'avenir' programme (ANR-10-IAHU-01) to the Imagine Institute. A.P. is a Centre National de la Recherche Scientifique Investigator. Open Access funding provided by Agence Nationale de la Recherche. Deposited in PMC for immediate release.

## Data and resource availability

The Shiny App to explore scRNAseq data from the mouse septum is available at: https://apps.institutimagine.org/mouse_septum/. Additionally, the Shiny App that allows making the pseudotime reconstructions of the ventral pallium (Moreau et al., 2021) is available at: https://apps.institutimagine.org/mouse_pallium/. Raw scRNAseq reads and processed count matrix are available from GEO (accession number GSE229603). The barcodes, SPRING coordinates and metadata of cells retained after quality control, as well as annotated R codes, have been deposited at https://fcauseret.github.io/septum/. All other relevant data and details of resources can be found within the article and its supplementary information.

## Peer review history

The peer review history is available online at https://journals.biologists.com/dev/lookup/doi/10.1242/dev.205011.reviewer-comments.pdf

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
