## [Peer Review File · Development (Cambridge, England)]

Bidirectional interaction between protocadherin 8 and the transcription factor Dbx1 regulates cerebral cortex development

Andrzej W. Cwetsch, Sofia Ferreira, Elodie Delberghe, Javier Gilabert-Juan, Matthieu X. Moreau, Yoann Saillour, Pau García-Bolufer, Saray Calvo-Parra, Jose González-Martínez, Durcia Massoukou, Ugo Borello, Frédéric Causeret and Alessandra Pierani
DOI: 10.1242/dev.205011

Editor: Debra Silver

Review timeline

Original submission:	7 June 2025
Editorial decision:	8 August 2025
First revision received:	10 November 2025
Accepted:	30 November 2025

Original submission

First decision letter

MS ID#: dev.205011

MS TITLE: Bidirectional interaction between Protocadherin 8 and transcription factor Dbx1 regulates cerebral cortex development

AUTHORS: Alessandra Pierani, Andrzej W. Cwetsch, Sofia Ferreira, Elodie Delberghe, Javier Gilabert-Juan, Matthieu X. Moreau, Yoann Saillour, Pau Garcia Bolufer, Saray Calvo Parra, Jose González Martínez, Durcia Massoukou, Ugo Borello and Frédéric Causeret

Dear Dr Pierani,

I have now received all the referees' reports on the above manuscript, and have reached a decision. The referees' comments are appended below, or you can access them online: please go to: *****

As you will see, the referees express considerable interest in your work, but have some significant criticisms and recommend a substantial revision of your manuscript before we can consider publication. If you are able to revise the manuscript along the lines suggested, which will involve some additional experiments, I will be happy receive a revised version of the manuscript. The reviewers make several suggestions regarding towards manuscript organization, necessary data analysis and suggest some additional markers/experiments to support conclusions. Please note you can also make use of the "limitations" section for addressing points which may be appropriate. Your revised paper will be re-reviewed by one or more of the original referees, and acceptance of your manuscript will depend on your addressing satisfactorily the reviewers' major concerns. Please also note that Development will normally permit only one round of major revision. If it would be helpful, you are welcome to contact us to discuss your revision in greater detail. Please send us a point-by-point response indicating your plans for addressing the referees' comments, and we will look over this and provide further guidance.

Please attend to all of the reviewers' comments and ensure that you clearly highlight all changes made in the revised manuscript. Please avoid using 'Tracked changes' in Word files as these are lost in PDF conversion. I should be grateful if you would also provide a point-by-point response detailing

how you have dealt with the points raised by the reviewers in the 'Response to Reviewers' box. If you do not agree with any of their criticisms or suggestions please explain clearly why this is so

Reviewer 1

Advance summary and potential significance to field

In the manuscript "Bidirectional interaction between Protocadherin 8 and transcription factor Dbx1 regulates cerebral cortex development", Cwetsch et al. investigate the interplay between the transcription factor Dbx1 and the cell adhesion molecule Pcdh8 in mouse cortical development. The authors found the temporally restricted ectopic expression (EE) of Dbx1 in the lateral pallium (LP) could induce cell aggregation and modify neuronal identity specifically. This process requires Pcdh8, which is induced by Dbx1. Interestingly, the authors further demonstrated that the ectopic expression of Pcdh8 could also induce Dbx1 expression and alter cortical organization changes. At the molecular level, they showed that Notch signaling is a downstream pathway for Pcdh8.

This is an interesting study highlighting a novel bidirectional regulatory loop between Dbx1 and Pcdh8 in cortical development. The data are compelling. I do have a few concerns as following.

General points:

1. While the Dbx1 and Pcdh8 EE results are very interesting, the manuscript would benefit from a more extensive discussion of the potential endogenous function of the Dbx1-Pcdh8 genetic loop in cortical development. Given the highly specific expression of Dbx1 (and Pcdh8) in ventral pallium (VP) progenitors just above the pallial-subpallial boundary (PSB) and the septum, it is plausible that the Dbx1-Pcdh8 bidirectional regulation plays a role in segregating VP and septum progenitor domain from adjacent cortical territories and/or regulating the migration of neurons derived from these regions. Incorporating this perspective could strengthen the significance of the findings and link the EE results to in vivo developmental mechanisms.
2. The conclusion that neurons derived from either Dbx1 EE or the Dbx1 progenitors are CLA neurons needs stronger support. In this study, the authors defined CLA neurons based on the expression of Nr4a2 and the absence of Ctjp2. However, these features are not specific to CLA neurons. To support this claim, additional CLA markers should be included to validate the identity of Dbx1 derived neurons. Further, to support the role of Dbx1 in generating CLA neurons, the authors could examine whether the generation of CLA neurons is disrupted in Dbx1 KO animals.
3. Although Dbx1 and Pcdh8 can induce each other's expression, their EE leads to distinct phenotypes. Specifically, while Dbx1EE induces robust cell clustering, which depends on Pcdh8, Pcdh8 EE alone could not induce clustering, despite its ability to upregulate Dbx1. This suggests that other mechanisms, such as their relative expression levels, timing, or cell types, might influence the phenotypic outcome of this genetic loop. The authors should discuss these possible context-dependent mechanism to reconcile the different effects in Dbx1EE and Pcdh8EE conditions.

Specific points:

1. Comments on Figure 1.
 - a. In Figure 1D, the authors quantify the number of cell clusters in MP, DP, and LP to show that Dbx1EE-induced clustering is region-specific. However, since IUE efficiency can vary between samples and regions, the absolute number of clusters may not reflect true regional preference. To account for this variability, the number of cell clusters should be normalized to the density of GFP+ cells in each region.
 - b. The current use of a line graph in Figure 1D suggests a continuous gradient from medial to lateral, which may be misleading. A bar graph would more clearly depict LP-specific cluster formation.
 - c. The rationale for the cell adhesion assay, demonstrating increased adhesion of Dbx1-expressing cells to N-cadherin-expressing cells, is unclear. The authors should clarify the rationale of this experiment and how it relates to the proposed role of Dbx1 in regulating cell aggregation via Pcdh8 or other adhesion molecules.
 - d. In Figure 1I, the GFP signals appear overexposed, particularly in the Dbx1EE panels, making it difficult to assess co-localization with Nr4a2. Improved image presentation at lower exposure would help to visualize co-expression patterns more accurately.

2. Comments on Figures 2, S3,
 - a. In Figures 2D-E, the "smoothed expression profiles" shown as curves are not clearly defined. It would be helpful if the authors clarified whether these curves represent the average reads among individual APs, or other measurements of gene expression levels. Providing details about the computational method used would aid in data interpretation.
 - b. The observation that *Pcdh8* is enriched in APs while *Dbx1* is enriched in BPs in VP is intriguing (Figure 2E). Given that *Pcdh8* expression is reduced in *Dbx1* KO animals (Figure S3C), it would be informative to determine whether this reduction is preferentially occurring in BPs, which would support the cell autonomous role of *Dbx1* in inducing *Pcdh8* expression.
 - c. Figure 2F (also Figure S3), it would be helpful to know whether the pallium cells be further clustered into MP, LP, and VP subgroups. It would be interesting to see whether *Dbx1*⁺ *Pcdh8*⁺ cells are within the VP cluster.
3. Comments on Figures 3, S6.
 - a. Discrepancies between qPCR and correlation analysis for *Dbx1* and *Pcdh8* should be better explained. For example, *Pcdh19* expression was not significantly different between control and *Dbx1*EE based on qPCR analyses (Figure 3), but it showed negative correlation in Figure S5A. While *Pcdh8* is significantly upregulated in the *Dbx1* EE but did not show positive correlation. The authors should explain the different methods used to analyze these results.
 - b. The E18 data was complicated, as the authors commented. The differences between control and *Dbx1*EE are not easy to observe. For example, the GFP⁺ cells in control also show high level of co-localization with *Pcdh8*⁺ (Figure S6A-b), similar to that in *Dbx1* EE (Figure S6A-d). It is possible it is due to the overexposed immunostaining signals. Higher magnification with better exposure might allow the readers to detect the differences.
 - c. The finding that *Dbx1*EE downregulates *Pcdh19* is interesting. A double immunostaining of *Pcdh8* and *Pcdh19* in the same sections could demonstrate whether *Dbx1*EE switches the cell adhesion molecule expression from *Pcdh19* to *Pcdh8*, potentially enabling cell clustering. This would provide compelling evidence for the role of *Dbx1* in switching protocadherin expression.
 - d. In Figure S6B, the *Pcdh19* expression seems to be relatively prominent, which seems inconsistent with the low *Pcdh19* expression in LP after E12.5 shown by in situ hybridization (Figure S5 G-I). Since the authors propose that low *Pcdh19* expression in LP permits *Dbx1*EE-induced clustering, it is important to verify that the *Pcdh19* signals in S6B are specific and not due to overexposure.
4. Comments on Figures 4 and S7,
 - a. The lack of an observable phenotype following *Pcdh8* KD in the LP is likely due to minimal endogenous *Pcdh8* expression in this region. It would be informative to test whether *Pcdh8* KD has a stronger effect in the VP, where its expression is normally higher.
 - b. In addition to compare *Nr4a2* and *Ctip2* expression percentages, the authors should also compare the cluster numbers or cluster forming efficiency between *Dbx1*EE and *Dbx1*EE+ *Pcdh8* KD. This comparison would strengthen the conclusion that *Pcdh8* mediates the aggregation phenotype induced by *Dbx1* EE.
5. Comments on Figures 5-7, S8
 - a. The image presentation across these figures is not fully consistent. While rosette-like structures induced by *Pcdh8*EE are clearly visible in Figure 5, they are less easy to observe in Figure 6, making it difficult to assess how consistent the phenotype is reproduced. To improve clarity and consistency, the authors could consider outlining the rosette-like structures in all relevant figures to ensure that key features are comparably presented.

Minor Comments

1. Better figure citation throughout the manuscript would benefit the readers. For example, pg5,: "When embryos were collected 141 7 days after IUE (E11.5-18.5), *Dbx1* EE⁺/*Nr4a2*⁺ cells accumulated in the intermediate zone (IZ)/SP (Figure 1I "d"). Within the CP, *Dbx1* EE⁺ were *Nr4a2*⁻ and coexisted with *Nr4a2*⁺ non-electroporated cells (Figure 1I "c")." could be revised for clarity.
2. Figure 3G - The boxed region shown should be shifted slightly dorsally.
3. Figure S5G-I - It would be informative to present *Pcdh* gene expression side-by-side with *Dbx1* expression to allow readers to compare the temporal and spatial expression patterns of these genes.
4. In Fig. 4B-C, it would be informative to include *Dbx1* immunostaining of a control (either *Dbx1* EE or *Dbx1*EE+control shRNA) samples to help visualize impact of *Pcdh8* knockdown on *Dbx1* protein expression.

Reviewer 2*Advance summary and potential significance to field*

1. Overall Evaluation

This manuscript investigates the bidirectional interaction between Protocadherin 8 (Pcdh8) and the transcription factor Dbx1 and its role in regulating cell fate determination and tissue patterning during cerebral cortex development. Traditionally, the relationship between transcription factors and cell adhesion molecules has been considered unidirectional. However, this study provides a novel perspective by demonstrating that Pcdh8 regulates Dbx1 expression, influencing cell fate and tissue structure via Notch signaling. The use of advanced techniques such as single-cell RNA sequencing (scRNAseq) and in utero electroporation (IUE) to dissect molecular mechanisms is highly commendable. In particular, the finding that Pcdh8 ectopic expression (EE) induces Dbx1 expression and promotes claustrum-amygdalar complex (CLA) neuron generation highlights a novel role for cell adhesion molecules in neural development. Furthermore, the demonstration of Notch ligand Jag1 involvement through rescue experiments significantly contributes to elucidating the molecular mechanisms.

However, several points require clarification or additional details, and there are some concerns regarding data interpretation and experimental design. Below, comments are organized by major sections.

Comments for the author

2. Introduction

Line 95-98: The preceding text cites prior studies showing that Dbx1 expression in the dorsal pallium (DP) can convert cell fate to Cajal-Retzius (CR) cells or subplate (SP)-like neurons. Therefore, the description of the current experiment starting from Line 95 is inaccurate without specifying that the results for CLA neurons pertain to cases where Dbx1 is introduced into the lateral pallium (LP).

Suggestion: Add a phrase such as "when introduced into the LP" after mentioning CLA to ensure clarity and accuracy.

3. Results

1. Section Title: "Ectopic Dbx1 expression in the lateral pallium induces cell aggregation"

The first half of this section appropriately discusses cell aggregation as indicated by the title. However, the latter half (starting around Line 129) shifts to describing the molecular expression characteristics of aggregated cells, which does not align with the section title.

Suggestion: Split this section into two, starting a new section around Line 129 with a revised title to better reflect the content, such as "Molecular Characteristics of Dbx1-Induced Aggregates."

2. Figure 1: The reason why Dbx1 EE-induced cell aggregation is specific to the LP is not sufficiently explained. Further discussion is needed on how the roles of other protocadherins (e.g., Pcdh9, Pcdh19) or regional differences in expression patterns contribute to this specificity.

Suggestion: Provide a more detailed explanation of the regional specificity of aggregation, potentially referencing Figure S5, in the Results or Discussion section.

3. Line 141-145: The intent of this section is unclear. Why was it necessary to examine embryos seven days after IUE? The significance of this timeframe is not well communicated.

Suggestion: Rewrite this section to clearly articulate the purpose of the seven-day observation and its relevance to the study's objectives.

4. Line 141-145 (CLA Neuron Markers): The manuscript states that Nr4a2+, Ctip2-, and Tle4- cells are characteristic of CLA neurons, but no references are provided to support this claim.

Suggestion: Cite relevant literature to substantiate the claim that these markers define CLA neurons.

5. Line 60-65: The description suggests a direct link between aggregation and the conversion to CLA neuron fate, which may be an overly strong assertion.

Suggestion: Move this discussion to the Discussion section to frame it as a hypothesis or interpretation rather than a definitive result, and provide a more nuanced explanation.

6. Figure 2A and 2D: The terminology used for brain regions (e.g., lateral anterior, lateral posterior) in Figure 2A differs from the MP, DP, and LP nomenclature used in Figure 1, which is

confusing. Similarly, the regional divisions in Figure 2D slightly differ from those in Figure 1, potentially causing reader confusion.

Suggestion: Add annotations in Figure 2A and 2D (e.g., "corresponds to DP in Figure 1") to clarify the relationship with Figure 1's nomenclature and reduce confusion.

7. Line 183-184: The sentence "We first used a previously generated dataset sampling cell diversity around the pallial-subpallial boundary (PSB) at E12.5" appears to be missing a reference to Figure 2D.

Suggestion: Add "(Figure 2D)" after this sentence to ensure proper referencing.

8. Line 228-277: The latter half of this section (starting around Line 259) contains interpretive content that is more suitable for the Discussion section. The Results section should focus on presenting factual findings.

Suggestion: Move the interpretive content from Line 259 onward to the Discussion section to maintain clarity in the Results.

9. Line 307-308: The manuscript states that Pcdh8 overexpression causes cortical thickening, but Figure 5A suggests that this thickening is specific to the electroporated region.

Suggestion: Clarify whether the thickening is specific to the electroporated area and, if so, explicitly state this in the text.

4. Discussion

1. Insufficient Mechanistic Depth: The molecular mechanisms by which Pcdh8 regulates Dbx1 expression (e.g., signaling pathways, post-transcriptional regulation) are not sufficiently discussed. For instance, the potential interaction of Pcdh8's intracellular domain (ICD) with specific kinases or transcriptional co-factors requires further exploration.

Suggestion: Based on existing literature, propose specific hypotheses regarding Pcdh8's ICD-mediated signaling and suggest future experimental designs to test these hypotheses.

2. Lack of Discussion on Limitations: The manuscript does not adequately address experimental design limitations, such as the regional specificity of IUE or potential off-target effects of Pcdh8 knockdown (KD). This makes it unclear how confident we can be in the interpretation of the results.

Suggestion: Explicitly describe the limitations of the experimental design and their potential impact on result interpretation.

3. Clinical Relevance: The manuscript mentions that Pcdh8 mutations are associated with neurodevelopmental disorders like autism and schizophrenia, but it lacks a detailed discussion of how these findings relate to the current study.

Suggestion: Discuss how the results (e.g., CLA neuron abnormalities) might contribute to understanding behavioral deficits in neurodevelopmental disorders and their clinical significance.

4. Data Interpretation: As noted in the Results section, the reason why Dbx1 EE-induced cell aggregation is specific to LP/VP is not sufficiently explained. The roles of other protocadherins (e.g., Pcdh9, Pcdh19) and regional expression pattern differences need further discussion.

Suggestion: Expand the discussion on how Pcdh8 and other protocadherins contribute to regional specificity, referencing Figure S5 if applicable.

5. Control Experiments: The specificity of Pcdh8 KD and EE experiments is supported by limited control experiments (e.g., non-specific shRNA or EE of other protocadherins). The claim that Pcdh8 KD suppresses Dbx1 expression via post-transcriptional regulation (page 11, Figure 4) lacks supporting evidence from additional experiments (e.g., mRNA stability analysis or proteomics).

Suggestion: Include control experiments with non-specific shRNA or other protocadherin KD in supplementary materials to confirm Pcdh8 KD specificity. Additionally, propose or include experiments to validate the post-transcriptional regulation mechanism.

5. Methods

1. Statistical Analysis Details:

- The normalization method for qPCR is unclear. The reference gene used for qPCR normalization (e.g., GAPDH or β -actin) should be specified.
- The sample size details for IUE data analysis are not clearly described. For example, in Figure 1E, it is stated that $n=6$, but it is unclear whether this represents six independent mother mice with one embryo sampled per mother or, for instance, two IUE experiments with three embryos sampled from each mother.

Suggestion: Specify the reference gene used for qPCR normalization and its rationale. Clearly state the number of mother mice and embryos used in IUE experiments for each figure.

2. Knockdown Experiment Methods:

- The selection process for shRNA target sequences is not described. The target sequences and their selection criteria should be detailed.
- It is unclear whether non-specific shRNA controls were used to confirm the specificity and efficiency of Pcdh8 KD. Additionally, information on the extent of protein-level reduction (e.g., via Western blot) is needed.

Suggestion: Provide details on the shRNA target sequences and their selection process. Include data from non-specific shRNA controls and Western blot results showing the extent of Pcdh8 protein reduction in supplementary materials.

6. Other Minor Comments

* Line 194: The reference to Figure 2D appears to be a mistake and should likely be Figure 2E.

Suggestion: Correct the reference to Figure 2E.

* Line 391: Adding "in the LP" after "E12.5" would improve clarity.

Suggestion: Insert "in the LP" to specify the region.

* Grammar: Some grammatical errors and unclear expressions are present, particularly in the latter half of the Discussion section, which feels verbose.

Suggestion: Have the manuscript proofread by a native English speaker to improve clarity and conciseness.

* Ethical Considerations: The description of ethical considerations for animal experiments is appropriate but lacks specific details, such as the ethics committee approval number or reference to specific guidelines.

Suggestion: Include the ethics committee approval number and reference to relevant guidelines to enhance transparency.

Reviewer 3

Advance summary and potential significance to field

Pierani et al. describe a bidirectional interaction between PCDH8 and DBX1 that regulates neural identity and downstream developmental processes. This is an exciting and significant finding as even if the regulation of cell adhesion molecules by transcription factors is well established; the opposite scenario, where the adhesion identities regulate specific expression of transcription factors has not been previously explored. This observation opens a new avenue in the field to enhance our understanding on the role that cell adhesion molecules play in determining cell identity.

Comments for the author

This is a well-written manuscript with clear and well-presented figures. The experimental approach is rigorous, and the conclusions are supported by the data presented. There are only few considerations that will enhance the readability and the clarity of the manuscript:

1. In general, the text will benefit from less abbreviations, as it becomes hard to follow. In particular, authors should consider not abbreviating cell autonomous as CA and non-cell autonomous as NCA, it is not necessary, and it really decreases readability. Authors should also consider removing other abbreviations like CR for Cajal-Retzius, CLA for claustroramygdaloid complex, etc. Some abbreviations are used 1-2 times, so they are not needed, like ncPCDHs, and finally some abbreviations are never spelled out, like RC (rostror-caudal?).

2. Authors should consider rearranging figures as the panels do not follow the order by which they are referred to in the text. To enhance clarity, figures should be numbered the way they appear (e.g. figure S5G-I is mentioned in the text before fig S3 and S4; FigS3C is mentioned before fig S3A), even the panels in the main figures are not mentioned in the text in order (fig 2G is mentioned before fig 2D,E,F).

3. Authors should include "data not shown" if it was used to draw conclusions (eg. "together with data showing Nr4a2+ Dbx1-derived neurons in a E14.5 dataset using Dbx1Cre;TauLacZ reporter (data not shown), these findings indicate that Dbx1 progenitors in the mouse VP originate pallial Nr4a2+ neurons that migrate ventrally, corresponding to neurons of the future claustramygdaloid complex (CLA) (Puelles et al., 2016a), and not SP neurons.")
4. Please provide a more comprehensive rationale and details about how cells were delineated in panels a-d of figure fig2H. This is very relevant as is the evidence provided to claim pcdh8 and dbx1 co-expression.
5. In figure 3D, there's no housekeeping gene for normalization, authors should include such gene to make sure the quantity/quality of the mRNA is comparable between groups/samples.
6. Graph in figure 4F is missing the Dbx1EE + control shRNA bar, this is particularly important as authors observed numerous GFP+Dbx1- cells in comparison to Dbx1 EE alone. This observation was explained by the fact that the plasmid used for Dbx1 EE is driven by a heterologous CAG promoter and thus Pcdh8 shRNA likely affects Dbx1 expression post-transcriptionally. If this is the case, then the control shRNA should have the same effect.

Minor

- Please replace insets c,d in figure 1I for better quality pictures.
- For Fig 5G is hard to see the lamination in the control and thus to observe the described delamination in the PCDH8 EE, please replace the pictures or add arrows to orient the reader.
- Figure 7 description should have its own title and be separated from the previous section.

First revision

Author response to reviewers' comments

POINT-BY-POINT RESPONSES

Reviewer #1

Reviewer 1: In the manuscript "Bidirectional interaction between Protocadherin 8 and transcription factor Dbx1 regulates cerebral cortex development", Cwetsch *et al.* investigate the interplay between the transcription factor Dbx1 and the cell adhesion molecule Pcdh8 in mouse cortical development. The authors found the temporally restricted ectopic expression (EE) of Dbx1 in the lateral pallium (LP) could induce cell aggregation and modify neuronal identity specifically. This process requires Pcdh8, which is induced by Dbx1. Interestingly, the authors further demonstrated that the ectopic expression of Pcdh8 could also induce Dbx1 expression and alter cortical organization changes. At the molecular level, they showed that Notch signaling is a downstream pathway for Pcdh8.

This is an interesting study highlighting a novel bidirectional regulatory loop between Dbx1 and Pcdh8 in cortical development. The data are compelling. I do have a few concerns as following.

We would like to thank this Reviewer for considering our work both interesting and novel.

General points:

1. While the Dbx1 and Pcdh8 EE results are very interesting, the manuscript would benefit from a more extensive discussion of the potential endogenous function of the Dbx1-Pcdh8 genetic loop in cortical development. Given the highly specific expression of Dbx1 (and Pcdh8) in ventral pallium (VP) progenitors just above the pallial-subpallial boundary (PSB) and the septum, it is plausible that the Dbx1-Pcdh8 bidirectional regulation plays a role in segregating VP and septum progenitor domain from adjacent cortical territories and/or regulating the migration of neurons derived from these regions.

Incorporating this perspective could strengthen the significance of the findings and link the EE results to in vivo developmental mechanisms.

Reviewer 1 brings up a very interesting point. In the revised manuscript, we have expanded the

Discussion to address the potential boundary-like roles and the bidirectional segregation functions of Dbx1 and Pcdh8 in the dorsoventral organization of pallial and subpallial domains and progenitor cell segregation during development (lines 391-408): “We propose that high Pcdh8 expression in Dbx1 EE- expressing cells, combined with low Pcdh19 levels preferentially in LP/VP progenitors, promoted strong homophilic interactions and aggregate formation in vivo. Conversely, higher co-expression of other δ - Pcdhs (i.e. Pcdh19) in the DP likely reduces aggregation due to heterotypic incompatibility. Together, we believe that the strong cell-segregative character of Pcdh8, combined with dynamic developmental changes in adhesion molecule combinatorial codes, regional (medial-dorsal-lateral axis) and progenitor-specific differences along the DV axis (i.e. VP and septum versus subpallium), modulate adhesion strength (Bisogni et al., 2018) and possibly neuronal migration, thereby contributing to the segregation of distinct neuronal populations. Moreover, variations in adhesion properties can play a crucial role in organising properly functioning brain areas. Pcdh8 EE resulted in the reorganization of multiple TF expression crucial for correct DV patterning (e.g. Shh, Dbx1, Lhx2), suggesting reciprocal regulation between adhesion molecules and transcriptional networks. Notably, during early stages of spinal cord development, specific cell adhesion molecules have been shown to follow the expression of TFs in selected progenitor domains (Tsai et al., 2020), ensuring robust tissue patterning. Altogether, our findings suggest that Dbx1 expression is both temporally and spatially restricted, and together with Pcdh8, can influence TF spatial organization, thereby contributing to proper cortical development”.

We agree with the reviewer that it is indeed an interesting question whether this could also influence the migration of neurons derived from these regions. This manuscript focuses on the interaction of Dbx1 and Pcdh8 in progenitors and early differentiation, but not on migration of EE neurons. We have revised a sentence in the Discussion (lines 395-400) to mention that migration, and not only early fate specification, could also be affected upon Dbx1/Pcdh8 EE. However, we are reluctant to speculate further, since we cannot precisely manipulate the concentration of adhesion molecules. In fact, the strong aggregation phenotype observed suggests and this could alter migration differently from physiological concentrations.

2. The conclusion that neurons derived from either Dbx1 EE or the Dbx1 progenitors are CLA neurons needs stronger support. In this study, the authors defined CLA neurons based on the expression of Nr4a2 and the absence of Ctip2. However, these features are not specific to CLA neurons. To support this claim, additional CLA markers should be included to validate the identity of Dbx1 derived neurons. Further, to support the role of Dbx1 in generating CLA neurons, the authors could examine whether the generation of CLA neurons is disrupted in Dbx1 KO animals.

We acknowledge the reviewer’s concern regarding whether Dbx1-derived neurons adopt a CLA fate. Our conclusions are primarily based on the expression of Nr4a2, but not Ctip2, and the spatial distribution of Dbx1-derived neurons in this ventral telencephalic regions using genetic tracing.

- 1) First, we would like to stress that we define the CLA as claustrum-amygdaloid complex, comprising the claustrum, basal/lateral amygdala, endopiriform nucleus (DEn) and piriform cortex (Puelles et al., 2016a) since they share a related gene-regulatory program making it difficult to this date to define the clear anatomical boundaries between these structures, particularly in the embryo. These structures are better defined and characterized at late embryonic and postnatal stages, which poses inherent limitations when tracing their origins early in development.
- 2) We believe our results clearly show that Dbx1 progenitors give rise to CLA-like neurons in the mouse, based on:
 - i) Lineage tracing with $Dbx1^{LacZ}$, $Dbx1^{Cre};Rosa26^{YFP}$ and $Dbx1^{Cre};Rosa26^{lacZ}$ at multiple embryonic (E12.5, E14.5 and E18.5) and postnatal ages (this manuscript; Moreau et al., Development 2021; Puelles, J. Chem. Neuroanat. 2016a) showing their location in ventral telencephalic regions in the early embryo and CLA in late embryo/postnatally.

ii) ScRNAseq and histological validation (at E12.5, E14.5 and P0-P2), showing developmental trajectories of ventrally located Dbx1-derived cells at early stages (possibly the future CLA) and E18.5 or postnatal in the CLA, which express Nr4a2. Importantly, we do not detect Dbx1-derived Nr4a2⁺ cells in the lateral cortex (e.g revised Fig. S3B, C of this manuscript).

3) We agree that the number of markers used to conclude that Dbx1 EE give rise to CLA neurons is limited and, without anatomical tracing, we cannot formally conclude that these are CLA neurons. That said, we believe the following arguments support our interpretation:

i) In the embryonic telencephalon, there are 3 main Nr4a2⁺ populations: subplate cells that are Ctip2⁺ (Arai *et al.*, *Cell Reports* 2019) and Bhlhe22⁻ (Supporting Fig. 1 panel a), caudolateral cortical plate neurons, especially caudal later-born pyramidal neurons (not Dbx1-derived in our tracing experiments; revised Fig. S3 of this manuscript) that are Ctip2⁺ (Supporting Fig. 2) and Bhlhe22⁺ (Supporting Fig. 1 panel b), and future CLA neurons located ventrally (Bhlhe22⁻) (Moreau *et al.*, *Development* 2021).

Supporting Fig. 1: Confocal images of a coronal section of E18.5 WT mouse brain, co-labeled with Bhlhe22 (green) and Nr4a2 (red). Dashed squares (a-b) are magnified on the right. Nr4a2⁺ subplate cells (a) do not express Bhlhe22. Arrowheads indicate Nr4a2⁺Bhlhe22⁺ neurons in the caudolateral cortical plate (b). Scale bars: 100 μ m, 25 μ m (magnified).

Supporting Fig. 2: Confocal images of a coronal section of E14.5 WT mouse brain, co-labeled with *Ctip2* (green), *Nr4a2* (red) and DAPI (blue) staining. Dashed squares (a) are magnified below. Arrowheads indicate *Nr4a2*⁺*Ctip2*⁺ neurons in the caudolateral cortical plate. Scale bars: 100 μ m, 50 μ m (magnified).

- ii) The *Dbx1* EE-induced *Nr4a2*⁺ neurons in the LP/VP at E13.5 are *Ctip2*⁻ (lines 123-127), thus excluding them as subplate neurons. In contrast, *Dbx1* EE in the dorsal pallium (DP) promotes *Ctip2*⁺ subplate-like neurons (Arai *et al.*, *Cell Reports* 2019).
- iii) At E13.5, the proportion of *GFP*⁺*Bhlhe22*⁺ upon *Dbx1* EE does not change compared to controls (revised Fig. S2E, F), indicating that the fate of caudolateral *Nr4a2*⁺ in the caudolateral cortex is not specifically induced by *Dbx1* EE.

Therefore, together these observations thus align with our interpretation that *Dbx1* EE-induced *Nr4a2*⁺ neurons display a molecular profile resembling ventrally located neurons with a CLA-like identity.

- 4) Finally, *Pcdh8* EE can significantly induce *Bhlhe22*⁺ neurons (revised Figs 5D, S8K of revised manuscript, and Supporting Fig. 3), while *Dbx1* EE and *shPcdh8* do not change this fate compared to control (Supporting Fig. 3). Therefore, we conclude that *Dbx1* EE does not induce *Nr4a2*⁺*Bhlhe22*⁺ neurons of the caudolateral cortical plate, consistent with lineage tracing data showing these neurons not being *Dbx1*-derived in the embryonic telencephalon (revised Fig. S3). The data shown in Supporting Fig. 3 has been included in the revised manuscript across two figures: revised Fig. S2E-F and Fig. S9C-D.

Supporting Fig. 3: (A) Confocal images of GFP (green) in coronal section of E13.5 mouse brain cortex electroporated at E11.5 with a control, *Pcdh8* EE, *Dbx1* EE or *Pcdh8* shRNA (*Pcdh8* KD) vector, co-labeled with *Bhlhe22* (red). Dashed squares highlighting GFP⁺ regions magnified on the right. Scale bar: 100 μ m. (B) Percentage of *Bhlhe22*⁺GFP⁺ over the total of GFP⁺ cells. Data are mean \pm SEM; circles represent values from independent electroporated embryos (n=3 controls, n=4 *Pcdh8* EE, n=3 *Dbx1* EE, n=3 *Pcdh8* shRNA). One-way ANOVA, post hoc Holm-Sidak, *p=0.0163.

While *Nr4a2* is the most typical CLA marker, excitatory neurons in this region have also been characterized by the expression of other enriched genes. However, most markers have been identified in adult mice and some very recently. There are currently no data describing developmental trajectories, making it very difficult to definitively match early embryonic subpopulations and their final location to firmly show what they become in the adult. To broaden our analysis, we have used recently published scRNAseq datasets on the CLA/DEN complex (Fodoulian *et al.*, *Nat Commun* 2025; Hara *et al.*, *Sci Rep* 2025; Kaur *et al.*, *Nature* 2025) to identify other markers of CLA neurons and neighbouring areas. We have identified several genes that are expressed in late embryonic and adult CLA (more than 10 CLA-enriched genes compared to adjacent regions included in the spatial transcriptomic data). Particularly, we have investigated the expression of *Lxn*, *Gnb4*, *Ntng2*, *Gng2*, *Oprk1*, *Fosl2*, *Egr2*, *Etl4*, *Smim32*, *Tfap2d*, *Lmo3*, *Stum* and *Scg2* in our scRNAseq datasets at E12, E12.5 (this study and Moreau *et al.*, 2021) and E14.5 (*Dbx1*-derived cells from *Dbx1*^{Cre};*Tau*^{GFP} mice; data not shown), and found either no expression in *Nr4a2*⁺ neurons or non-specific expression. These findings strongly suggest that CLA molecular identity consolidates only at later developmental stages, supporting that *Nr4a2* remains the most representative and specific marker for CLA neurons starting from early embryonic stages.

Lastly, both our scRNAseq analysis and histological studies with molecular markers indicate that *Dbx1*-derived CLA neurons represent small subset of the *Nr4a2*⁺ population varying along the antero-posterior (AP) axis, making it technically challenging to perform robust statistical analysis without other early markers, especially in the *Dbx1* KO embryos. We have carefully traced at E12.5 and E14.5 the ventral telencephalic *Nr4a2*⁺ populations (derived or not from *Dbx1* progenitors), and identified at these stages many subpopulations through coexpression of multiple markers (*Pbx3*, *Meis2*, *Etv1*, *Foxp2*, from Moreau *et al.*, 2021). However, not only these

populations are quite small in number and vary along the AP axis, but currently we lack specific markers for each subpopulation. We have additionally tested two of these candidate genes (*Pbx3* and *Meis2*) and found no expression in *Dbx1* EE GFP⁺ cells upon *Dbx1* EE (Supporting Fig. 4). This could reflect an incomplete maturation of CLA fate with the induced Nr4a2⁺ cells not being yet fully differentiated, and, although they persist at E18.5, they may not be able to migrate to their final destination (they are indeed found in the IZ). Alternatively, due to dynamic changes in molecular profiles over time, we may be just unable to reproduce the full developmental trajectory of these rare subpopulations and match embryonic to postnatal CLA populations.

Supporting Fig. 4: Confocal images of GFP (green) in coronal section of E13.5 mouse brain cortex electroporated at E11.5 with *Dbx1* EE vector, co-labeled with *Pbx3* (red), *Meis2* (white) and DAPI counterstaining. Dashed areas (a) and (b) highlighting *Dbx1EE*⁺ cells are magnified below.

We would like to acknowledge the limitations of these experiments. Most available datasets are derived from postnatal studies (e.g., Fodoulian *et al.*, 2025), whereas our experimental window focuses on embryonic development. Consequently, many markers described in the literature may not yet be expressed during the developmental stages we study. Furthermore, even if we identify additional markers in *Dbx1* KO embryos using in situ hybridization, the lack of suitable antibodies will limit experimental validation in in utero electroporation and quantitative studies, since it is the coexpression of multiple genes that characterize specific cell identities. Together, we believe that these limitations, although interesting and important to be considered, do not undermine the central focus of our study, which is the bidirectional interaction between transcription factors and cell adhesion molecules in cortical development. To avoid overstating our conclusion, we propose to replace “CLA neurons” with “CLA-like neurons” in the few sentences of the manuscript.

In the revised manuscript we explained the technical and developmental limitations in the new “Study limitations” section (lines 502-514) that prevent us from formally proving the identity of the induced Nr4a2⁺ neurons and whether they are fully differentiated or not.

3. Although *Dbx1* and *Pcdh8* can induce each other's expression, their EE leads to distinct phenotypes. Specifically, while *Dbx1EE* induces robust cell clustering, which depends on *Pcdh8*, *Pcdh8* EE alone could not induce clustering, despite its ability to upregulate *Dbx1*. This suggests

that other mechanisms, such as their relative expression levels, timing, or cell types, might influence the phenotypic outcome of this genetic loop. The authors should discuss these possible context-dependent mechanism to reconcile the different effects in Dbx1EE and Pcdh8EE conditions.

We agree with the reviewer that this is an important difference that should be explained in the Discussion section. We had provided the explanation on page 15, lines 457-462 of the Discussion of the original manuscript: “The observed differences likely derive from the timing of Pcdh8 expression, whether directly driven by the pCAGGS promoter/enhancer or induced later by Dbx1 EE (itself being first expressed under pCAGGS control). As Dbx1 promotes cell cycle exit (García-Moreno et al., 2018), pCAGGS-driven Pcdh8 will induce expression directly in APs, whereas pCAGGS-driven Dbx1 expression in APs will trigger rapid cell cycle exit, thus likely inducing Pcdh8 expression in young neurons (or BPs)”. We are confident the reviewer will agree with this explanation, now found in lines 455-461 of the revised manuscript.

Specific points:

1. Comments on Fig. 1.

a. In Fig. 1D, the authors quantify the number of cell clusters in MP, DP, and LP to show that Dbx1EE- induced clustering is region-specific. However, since IUE efficiency can vary between samples and regions, the absolute number of clusters may not reflect true regional preference. To account for this variability, the number of cell clusters should be normalized to the density of GFP⁺ cells in each region.

We applied a new approach for quantifying the formed clusters across the pallium (Supporting Fig. 6). For each electroporated section, MP, DP, and LP boundaries were defined consistently using identical ROI masks across all samples. Within each ROI, GFP⁺ cell density was calculated by dividing the total number of GFP⁺ cells by the ROI area. To obtain accurate GFP⁺ cell counts under Dbx1 EE conditions (where clustered cells are tightly packed and GFP signals overlap), we relied on nuclear DAPI staining to distinguish individual cells. Finally, the number of clusters (defined by more than 3 GFP⁺ cells grouped in close proximity) was normalized to the GFP⁺ cell density within each ROI.

The schematics illustrating this quantification (Supporting Fig. 5), along with the detailed description of the new analysis in the Materials and Methods section (lines 801-806) of the revised manuscript, has been included as revised Fig. S1D-E.

Supporting Fig. 5: (A) Representative images showing GFP (green) clusters and non-clusters (outlined with dashed lines) at E13.5 upon IUE of Dbx1 EE at E11.5 in mouse cortices. Individual GFP⁺ cells (dots) were identified based on nuclear DAPI (blue) counterstaining. (B) A

cluster was defined as a group of 3 or more GFP⁺ cells in close proximity. (C) The medial, dorsal and lateral pallium regions were consistently defined using identical ROI masks across all control and Dbx1 EE samples. GFP⁺ cell density was calculated by dividing the total number of GFP⁺ cells by the ROI area.

b. The current use of a line graph in Fig. 1D suggests a continuous gradient from medial to lateral, which may be misleading. A bar graph would more clearly depict LP-specific cluster formation.

We have modified the graph (revised Fig. 1D) in accordance with the reviewer's suggestion.

c. The rationale for the cell adhesion assay, demonstrating increased adhesion of Dbx1-expressing cells to N-cadherin-expressing cells, is unclear. The authors should clarify the rationale of this experiment and how it relates to the proposed role of Dbx1 in regulating cell aggregation via Pcdh8 or other adhesion molecules.

We have clarified the rationale for this experiment in the manuscript by indicating that it was designed as an initial step in the study when we detected clusters of GFP⁺ induced by Dbx1 EE, conducted prior to our knowledge of the involvement and upregulation of specific adhesion molecules. The primary aim was to use a well-established standard assay to determine whether Dbx1 EE cells exhibit enhanced adhesive properties in vitro, which could, in turn, underlie their capacity to form clusters in vivo. This has been added in lines 107-109: "Knowing that cell aggregation often relies on stronger cell-cell or cell-substrate adhesion, we next investigated whether Dbx1 EE affected cell adhesiveness. To this end, we performed a well-established *n* adhesion assay, as previously described (Porlan et al., 2014)".

d. In Fig. 1I, the GFP signals appear overexposed, particularly in the Dbx1EE panels, making it difficult to assess co-localization with Nr4a2. Improved image presentation at lower exposure would help to visualize co-expression patterns more accurately.

We appreciate the reviewer's concern regarding the apparent overexposure of the image. We have carefully optimized the image and provide an improved version of revised Fig. 1I.

2. Comments on Figs 2. S3,

a. In Figs 2D-E, the "smoothed expression profiles" shown as curves are not clearly defined. It would be helpful if the authors clarified whether these curves represent the average reads among individual APs, or other measurements of gene expression levels. Providing details about the computational method used would aid in data interpretation.

We have included an additional explanation in the Materials and Methods section (lines 895-898) stating that the curves represent normalized gene expression, calculated using the `NormalizeData()` function of Seurat with the parameters `normalization.method = "LogNormalize"` and `scale.factor = 10000`. The curves were plotted using the `geom_smooth()` function of ggplot2 with default parameters.

b. The observation that Pcdh8 is enriched in APs while Dbx1 is enriched in BPs in VP is intriguing (Fig. 2E). Given that Pcdh8 expression is reduced in Dbx1 KO animals (Fig. S3C), it would be informative to determine whether this reduction is preferentially occurring in BPs, which would support the cell autonomous role of Dbx1 in inducing Pcdh8 expression.

We observed changes in Pcdh8 mRNA expression within both the progenitor domains and the postmitotic compartment of Dbx1 KO embryos. We did not initially detailed what changes we observed since they are complex and involve cell autonomous- and possibly non-cell autonomous-related effects at the pallial-subpallial boundary (VP apical/basal progenitor domain) as well as in the postmitotic compartment. This complexity likely arises from the highly dynamic coexpression of Dbx1 and Pcdh8 during the AP/BP transition (revised Fig. 2F). We have described this section extensively in lines 182-226 of the Results in the original manuscript. Together with the observed reduction of Pcdh8⁺ neurons in the postmitotic compartment of the Dbx1 KO in

the revised Fig. S5D (white arrowheads), these observations suggest that *Dbx1* regulates the number of *Pcdh8* neurons generated. Given that *Dbx1* is known to be highly expressed in G1 and to promote cell-cycle exit (García-Moreno *et al.*, 2018), we believe the reduction of postmitotic *Pcdh8*⁺ neurons in the KO is due to the impairment of cell-cycle exit, preventing progenitors from differentiating into *Pcdh8*⁺ neurons. We have clarified this interpretation more explicitly in the Discussion section (lines 415-422): “We showed changes in *Pcdh8* expression domains in *Dbx1*^{LacZ/LacZ} KO mice, notably a reduction in the *Pcdh8*⁺ postmitotic compartment, suggesting that *Dbx1* regulates *Pcdh8* expression in a region-specific manner, restricting its expression in septal progenitors while promoting *Pcdh8*-expressing neuronal differentiation in the VP. This pattern correlates with the emergence of the lateral postmitotic *Pcdh8*⁺ (*Nr4a2*⁺) population; however, this population appears partially lost in the absence of *Dbx1*, likely resulting from impaired cell-cycle exit (García-Moreno *et al.*, 2018)”.

c. Fig. 2F (also Fig. S3), it would be helpful to know whether the pallium cells be further clustered into MP, LP, and VP subgroups. It would be interesting to see whether *Dbx1*⁺ *Pcdh8*⁺ cells are within the VP cluster.

We have previously shown that pallial apical progenitors do not form discrete clusters, but rather exhibit a continuum of cell identities along the dorso-to-ventral axis (Moreau *et al.*, *Development* 2021). *Dbx1* expression is restricted to the VP, starting in apical progenitors and peaking in basal/intermediate progenitors (Figs 5E, F and 7E in Moreau *et al.*, 2021, as well as revised Figs 2D, E, G and H-3d of this manuscript). Accordingly, *Dbx1*⁺/*Pcdh8*⁺ cells are localized within the VP. We have revised the main text (lines 183-187) to improve clarity: “To further investigate the co-expression of *Dbx1* and *Pcdh8* at earlier stages in the VP, we produced a scRNAseq dataset of the entire telencephalic vesicle at E11.5-E12 and detected *Dbx1*⁺*Pcdh8*⁺ cells at the AP-to-BP transition (Fig. 2F), most likely belonging to the VP lineage, as *Dbx1* expression is confined to progenitors in this region (Fig. 2G) (Moreau *et al.*, 2021).”

3. Comments on Figs 3, S6.

a. Discrepancies between qPCR and correlation analysis for *Dbx1* and *Pcdh8* should be better explained. For example, *Pcdh19* expression was not significantly different between control and *Dbx1*EE based on qPCR analyses (Fig. 3), but it showed negative correlation in Fig. S5A. While *Pcdh8* is significantly upregulated in the *Dbx1* EE but did not show positive correlation. The authors should explain the different methods used to analyze these results.

We thank the reviewer for pointing out our unclear description of these results in the text. The Fig. S5A (now revised Fig. S6A) shows that *Pcdh19* expression in the same dissected brain sample inversely correlated with *Dbx1* EE levels when electroporation was done at E11.5 and analysed 48h later.

Specifically, samples with higher *Dbx1* expression had lower *Pcdh19* levels, suggesting that *Pcdh19* decreases in a *Dbx1* dosage-dependent manner. However, the mean *Pcdh19* expression across all *Dbx1* EE samples (revised Fig. 3D, F) remained unchanged compared to controls. This reflects both the variability in *Dbx1* expression across electroporated samples, thus of *Pcdh19*, and the contribution of non-electroporated cells present in the samples that endogenously express *Pcdh19*. Consistent with this interpretation, *Pcdh19* downregulation upon *Dbx1* EE at E11.5 is also visible by immunofluorescence in revised Fig. S7B. Lastly, at E12.5, IUE of *Dbx1* EE resulted in *Pcdh19* upregulation (revised Fig. S6E), highlighting the importance of progenitor competence at different developmental stages. At E12.5, *Dbx1* can induce a different set of *Pcdh8* compared to E11.5, in particular, *Pcdh8* at E11.5 and *Pcdh9/Pcdh19* at E12.5 (revised Figs 3D, F, S6E). These temporal differences likely contribute to variations in aggregation capacity and cell fate allocation. As discussed on page 13-14, lines 396-398 of the original manuscript: “This correlates with the induction of distinct set of *Pcdh8* at E11.5 and *Pcdh9/Pcdh19* at E12.5, suggesting that progenitor cell competence varies over time, giving rise to daughter cells with different *Pcdh8* expression combinations on their surface”. Regarding *Pcdh8* expression level, we observed the opposite pattern. No correlation with *Dbx1* EE levels was found when examining individual electroporated sample (revised Fig. S6A), but

averaged across all *Dbx1* EE samples, *Pcdh8* was significantly upregulated following electroporation (revised Fig. 3D, F).

We have revised the manuscript and included in the Results section (lines 227-233): “Additionally, following *Dbx1* EE (E11.5-13.5), we observed a significant negative correlation between *Pcdh19* and *Dbx1* gene expression (Fig. S6A), such that samples with higher *Dbx1* expression exhibited lower *Pcdh19* levels, pointing to a *Dbx1*-dose dependent regulation and other possible partners in gene co-regulation networks. This relationship was not observed for other cadherins (i.e. *Pcdh8*, *Pcdh9* and *Ncad*). Consistently, *Pcdh19* expression on sections was detected in DP but not in LP/VP progenitors at E12.5-E13.5 (Fig. S4B-C)”.

In the Materials and Methods section (lines 771-776), we have also added: “The correlation plots in Fig. S6A depict gene expression levels relative to *Dbx1* expression in each electroporated sample, whereas the analyses in Figs 3C-F and S6D,E represent mean gene expression values, normalized to *Gapdh*, across all control and *Dbx1* EE samples.”

b. The E18 data was complicated, as the authors commented. The differences between control and *Dbx1*EE are not easy to observe. For example, the GFP⁺ cells in control also show high level of co-localization with *Pcdh8*⁺ (Fig. S6A-b), similar to that in *Dbx1* EE (Fig. S6A-d). It is possible it is due to the overexposed immunostaining signals. Higher magnification with better exposure might allow the readers to detect the differences.

The figure in question has been revised with improved exposure and higher magnification as suggested, and is now presented as Fig. S7A in the revised manuscript.

c. The finding that *Dbx1*EE downregulates *Pcdh19* is interesting. A double immunostaining of *Pcdh8* and *Pcdh19* in the same sections could demonstrate whether *Dbx1*EE switches the cell adhesion molecule expression from *Pcdh19* to *Pcdh8*, potentially enabling cell clustering. This would provide compelling evidence for the role of *Dbx1* in switching protocadherin expression.

The double labeling on the same section is unfortunately not possible because the *Pcdh8* antibody requires antigen retrieval, a procedure that is not compatible with the *Pcdh19* antibody. Nevertheless, we believe that the consisting staining patterns observed in revised Fig. S7A, where *Dbx1* EE cells show high *Pcdh8* expression, and in revised Fig. S7B, where *Dbx1* EE cells display low *Pcdh19* expression, strongly suggest a change in *Pcdh* expressions in GFP⁺ cells.

d. In Fig. S6B, the *Pcdh19* expression seems to be relatively prominent, which seems inconsistent with the low *Pcdh19* expression in LP after E12.5 shown by *in situ* hybridization (Fig. S5 G-I). Since the authors propose that low *Pcdh19* expression in LP permits *Dbx1*EE-induced clustering, it is important to verify that the *Pcdh19* signals in S6B are specific and not due to overexposure.

We appreciate the reviewer’s concern regarding the specificity of the antibody used in our study. This antibody has been previously validated and applied in both Western blot and immunohistochemical analyses in postnatal stages, as reported by Cwetsch et al. (Brain Communications, 2022). To further confirm its specificity in our experimental context, we generated a comparative figure (Supporting Fig. 6) showing *Pcdh19* mRNA *in situ* hybridization alongside *Pcdh19* immunostaining in mouse at E13.5. The reviewer can observe a strong correspondence between the two detection methods, with yellow arrowheads indicating regions of low signal and blue arrowheads marking identical zones of strong expression in both images. This overlap supports the specificity and reliability of the antibody used in our analyses. *Pcdh8* expression is very dynamic and, at E13.5, *Pcdh19* is highly expressed in the postmitotic compartment, perfectly correlating mRNA expression with protein expression (revised Fig. S7B). In addition, analysis of our scRNAseq datasets at E12, E12.5 and E14.5 (not shown) clearly indicate the increasing widespread expression of *Pcdh19* across both mitotic and postmitotic compartments.

Supporting Fig. 6: *Pcdh19* expression detected by in situ hybridization (ISH) (top) and immunofluorescence (bottom) in the WT developing mouse brain at E13.5. Regions showing high or low signal intensity are indicated by blue and yellow arrowheads, respectively. Scale bars: 200 μm .

4. Comments on Figs 4 and S7,

a. The lack of an observable phenotype following *Pcdh8* KD in the LP is likely due to minimal endogenous *Pcdh8* expression in this region. It would be informative to test whether *Pcdh8* KD has a stronger effect in the VP, where its expression is normally higher.

We have replaced the image of *Pcdh8* KD condition in revised Fig. S8D with another illustrating a more ventrally localized electroporation. As the reviewer can observe, this modification does not alter the phenotype in more VP regions, and the transfected cells continue to exhibit a control-like identity.

b. In addition to compare *Nr4a2* and *Ctip2* expression percentages, the authors should also compare the cluster numbers or cluster forming efficiency between *Dbx1*EE and *Dbx1*EE+*Pcdh8* KD. This comparison would strengthen the conclusion that *Pcdh8* mediates the aggregation phenotype induced by *Dbx1* EE.

We appreciate the reviewer's suggestion to include the analysis illustrating the rescue of the cluster formation phenotype in the *Dbx1* EE + *Pcdh8* shRNA condition. We show here an additional graph (Supporting Fig. 7) showing that cell clusters are no longer formed under the *Dbx1* EE + *Pcdh8* KD condition compared with *Dbx1* EE + control shRNA (scramble) and *Dbx1* EE alone. To ensure consistency and comparability across conditions, we focused our quantification on the

lateral pallium, where the phenotype was most pronounced and where representative samples were available for all experimental groups, as not all electroporations encompassed medial or dorsal regions. These additional data has been included in the revised manuscript as **revised Fig. S8G**.

Moreover, the lack of cluster formation upon Pcdh8 KD can be appreciated from the comparison of GFP images in Dbx1 EE + Pcdh8 KD (**revised Fig. 4C**) with the GFP⁺ clusters observed in Dbx1 EE + control shRNA (**revised Fig. 4B**) and Dbx1 EE alone (e.g. **revised Figs 1B, 3G, S1B**).

Supporting Fig. 7: Pcdh8 KD in Dbx1 EE prevents cell cluster formation. Data are mean \pm SEM; circles represent single analysed section values ($n=6$ controls, $n=6$ Dbx1EE, $n=4$ Dbx1 EE + control shRNA, $n=4$ Dbx1 EE + Pcdh8 KD) from independent electroporations (from at least 3 animals). One-way ANOVA, post hoc Holm-Sidak: ns, not significant; *** $p < 0.001$.

5. Comments on Figs 5-7, S8

a. The image presentation across these Figs is not fully consistent. While rosette-like structures induced by Pcdh8EE are clearly visible in Fig. 5, they are less easy to observe in Fig. 6, making it difficult to assess how consistent the phenotype is reproduced. To improve clarity and consistency, the authors could consider outlining the rosette-like structures in all relevant Figs to ensure that key features are comparably presented.

The rosette-like structures have been outlined in **revised Fig. 6** for better clarity and interpretability.

Minor Comments

1. Better Fig. citation throughout the manuscript would benefit the readers. For example, pg5,: "When embryos were collected 141 7 days after IUE (E11.5-18.5), Dbx1 EE⁺/Nr4a2⁺ cells accumulated in the intermediate zone (IZ)/SP (Fig. 1I "d"). Within the CP, Dbx1 EE⁺ were Nr4a2⁻ and coexisted with Nr4a2⁺ non-electroporated cells (Fig. 1I "c")." could be revised for clarity.

The figures have been referenced more clearly throughout the manuscript to improve clarity and enhance readability.

2. Fig. 3G - The boxed region shown should be shifted slightly dorsally.

The annotation has been corrected in **revised Fig. 3G**.

3. Fig. S5G-I - It would be informative to present Pcdh gene expression side-by-side with Dbx1 expression to allow readers to compare the temporal and spatial expression patterns of these genes.

We thank the reviewer for this valuable comment and fully understand the rationale for including an additional *Dbx1* *in situ* hybridization. To address this point, we have incorporated *Dbx1* expression patterns into the schematic diagrams (**revised Fig. S4C**), following the approach by Bielle *et al.* (Bielle *et al.*, *Nat Neurosci* 2005) for E11.5, and for E12.5 as shown in **revised Fig. 2G** of this manuscript. The remaining E13.5 timepoint is provided below for reviewers only (**Supporting Fig. 8**). We do not have sections that correspond precisely to those used for the *Pcdh* analyses presented in the **revised Fig. S4A, B**. Therefore, a direct one-to-one comparison is not feasible. Nonetheless, we believe that overlaying the well-established *Dbx1* expression domains onto the schematic representation (**revised Fig. S4C**) provides a clear and accurate framework to illustrate the relevant correlations.

Supporting Fig. 8: Representative bright-field images of *in situ* hybridization for *Dbx1* in coronal brain sections of E13.5 wild-type embryo. DP, dorsal pallium; LP, lateral pallium; VP, ventral pallium. Scale bars: 100 μ m.

4. In Fig. 4B-C, it would be informative to include *Dbx1* immunostaining of a control (either *Dbx1* EE or *Dbx1*EE+control shRNA) samples to help visualize impact of *Pcdh8* knockdown on *Dbx1* protein expression.

We thank the reviewer for this suggestion and understand the concern. However, the requested information is already included in the manuscript, presented across two figures: **revised Fig. S1B,C**, which shows *Dbx1* EE-induced *Dbx1* protein expression, and **revised Fig. 4C**, which demonstrates the absence of *Dbx1* protein expression under the *Dbx1* EE + *Pcdh8* shRNA condition. Additionally, we have **revised Fig. 4F** to include quantification of *Dbx1* expression across all analysed conditions. We believe that reorganizing these results into a single panel would disrupt the logical flow of the manuscript and reduce the clarity of data presentation.

Reviewer #2

Reviewer 2: SUMMARY OF THE ADVANCE MADE IN THIS PAPER AND ITS POTENTIAL SIGNIFICANCE TO THE FIELD

1. Overall Evaluation

This manuscript investigates the bidirectional interaction between Protocadherin 8 (*Pcdh8*) and the transcription factor *Dbx1* and its role in regulating cell fate determination and tissue patterning during cerebral cortex development. Traditionally, the relationship between transcription factors and cell adhesion molecules has been considered unidirectional. However, this study provides a novel perspective by demonstrating that *Pcdh8* regulates *Dbx1* expression, influencing cell fate and tissue structure via Notch signaling. The use of advanced techniques such as single-cell RNA sequencing (scRNAseq) and *in utero* electroporation (IUE) to dissect molecular mechanisms is highly commendable. In particular, the finding that *Pcdh8* ectopic expression (EE) induces *Dbx1* expression and promotes claustrum-amygdalar complex (CLA) neuron generation highlights a novel role for cell adhesion molecules in neural development. Furthermore, the demonstration of Notch ligand *Jag1* involvement through rescue experiments significantly contributes to elucidating the molecular mechanisms.

However, several points require clarification or additional details, and there are some concerns regarding data interpretation and experimental design. Below, comments are organized by major sections.

We are delighted that the reviewer considers our study to offer a novel perspective and finds it highly commendable. We have thoroughly addressed all comments to enhance clarity of the manuscript and fulfil the reviewer's expectations.

SUGGESTIONS TO AUTHORS

2. Introduction

Line 95-98: The preceding text cites prior studies showing that Dbx1 expression in the dorsal pallium (DP) can convert cell fate to Cajal-Retzius (CR) cells or subplate (SP)-like neurons. Therefore, the description of the current experiment starting from Line 95 is inaccurate without specifying that the results for CLA neurons pertain to cases where Dbx1 is introduced into the lateral pallium (LP). Suggestion: Add a phrase such as "when introduced into the LP" after mentioning CLA to ensure clarity and accuracy.

We have rephrased the sentence in accordance with the reviewer's suggestion.

3. Results

1. Section Title: "Ectopic Dbx1 expression in the lateral pallium induces cell aggregation"

The first half of this section appropriately discusses cell aggregation as indicated by the title. However, the latter half (starting around Line 129) shifts to describing the molecular expression characteristics of aggregated cells, which does not align with the section title.

Suggestion: Split this section into two, starting a new section around Line 129 with a revised title to better reflect the content, such as "Molecular Characteristics of Dbx1-Induced Aggregates."

We have followed the reviewer's suggestion and reorganized it in two sections: "*Ectopic Dbx1 expression in the lateral/ventral pallium induces cell aggregation*" (line 93), and "*Ectopic Dbx1 expression in the lateral/ventral pallium induces Nr4a2⁺ neurons resembling claustrum-amygdaloid complex identity*" (line 117).

2. Fig. 1: The reason why Dbx1 EE-induced cell aggregation is specific to the LP is not sufficiently explained. Further discussion is needed on how the roles of other protocadherins (e.g., Pcdh9, Pcdh19) or regional differences in expression patterns contribute to this specificity.

Suggestion: Provide a more detailed explanation of the regional specificity of aggregation, potentially referencing Fig. S5, in the Results or Discussion section.

We have elaborated on this aspect in the manuscript, particularly regarding the cell-segregative nature of the Pcdhs. It has been shown that δ -Pcdh (i.e. Pcdh8, Pcdh9 and Pcdh19) avoid to form aggregates with each other and exclusively mediate homophilic aggregation (Bisogni *et al.*, 2018). Notably, in the same study authors showed that cells expressing a single δ -Pcdh (i.e. Pcdh8), when mixed with another population expressing the same δ -Pcdh (i.e. Pcdh8) plus an additional 'mismatched' δ -Pcdh (i.e. Pcdh9), still strongly avoid aggregation. This suggests that adhesive affinity and relative surface expression levels can indeed regulate the co-aggregation behaviors of the cells. Our results show that expression levels of different δ -Pcdhs is distinct in LP versus DP (revised Fig. S4A-C). For example, at E11.5/E12, Pcdh9 exhibits strong expression in the postmitotic compartment with a lateral^{high}-to-medial^{low} gradient and is highly expressed in progenitors around the PSB, whereas at E12.5, Pcdh19 is strongly expressed in progenitors following a medial^{high}-to-lateral^{low} gradient, and in postmitotic neurons in the future piriform cortex. Thus, we speculate that a similar situation takes place when we induce high Pcdh8 expression upon Dbx1 EE in LP progenitors. LP progenitors will mediate stronger aggregation than DP progenitors or the postmitotic compartment, where high co-expression levels of other δ -Pcdhs, i.e. Pcdh19, will reduced the co-aggregation index (decrease aggregation). Consistently, the almost mutually exclusive low levels of Pcdh19 protein in the Pcdh8-expressing aggregates can be appreciated also in revised Fig. S7B. Moreover, the patterns of expression for Pcdh8 and other Pcdhs (discussed in the manuscript) vary extremely dynamically between E11.5 (IUE timepoint) and

E13.5 (analysis timepoint) as shown by in situ hybridization (**revised Fig. S4A-C**). The reviewer can appreciate developmental changes in adhesion molecule combinatorial code that are both cell- and region-specific (also visible in **revised Fig. 2B, C**), influencing adhesion strength, as previously shown (Bisogni *et al.*, 2018). We believe that the strong cell-segregative character of Pcdh8 and the quick changes in Pcdhs expression patterns along the DV axis likely explains the differences in aggregate formation between the dorsal and lateral pallium. Overall, we concluded that high levels of Pcdh8 in cells transfected with Dbx1 EE plasmid (1) enhances adhesive properties of the cells *in vitro* (**revised Fig. S1G-I**) and (2) promotes strong homophilic interactions (aggregate formation) *in vivo* that occur preferentially in progenitor cells with low levels of other δ -Pcdh (i.e. Pcdh19) in the LP (**revised Fig. S4A-C**). These data are described and discussed in the original manuscript, lines 243-244, 259-275 and 402-407.

We have revised and expanded the Discussion section (lines 383-403) on expression patterns and their potential role in driving aggregate formation: *“Consistently, our expression analyses show that δ -Pcdh (Pcdh8, Pcdh9, Pcdh19) display sharply distinct and dynamic spatial patterns between E11.5 and E13.5. Importantly, δ -Pcdhs mediate strictly homophilic adhesion and avoid heterotypic interactions (Bisogni *et al.*, 2018). Even subtle mismatches in δ -Pcdh expression can prevent co-aggregation, implying that adhesive affinity and surface composition determine segregation behaviour. This principle likely explains why Dbx1 EE in the LP leads transfected cells into Pcdh8-expressing aggregates with CLA-like neuron profile ($Nr4a2^+ Ctip2^- Bhlhe22^- Reln^- Calb2^-$), whereas in the DP, it did not induce aggregation but instead correlated with Cajal-Retzius and subplate-like neuron fates (Arai *et al.*, 2019). We propose that high Pcdh8 expression in Dbx1 EE-expressing cells, combined with low Pcdh19 levels preferentially in LP/VP progenitors, promoted strong homophilic interactions and aggregate formation *in vivo*. Conversely, higher co-expression of other δ -Pcdhs (i.e. Pcdh19) in the DP likely reduces aggregation due to heterotypic incompatibility. Together, we believe that the strong cell-segregative character of Pcdh8, combined with dynamic developmental changes in adhesion molecule combinatorial codes, regional (medial-dorsal-lateral axis) and progenitor-specific differences along the DV axis (i.e. VP and septum versus subpallium), modulate adhesion strength (Bisogni *et al.*, 2018) and possibly neuronal migration, thereby contributing to the segregation of distinct neuronal populations. Moreover, variations in adhesion properties can play a crucial role in organising properly functioning brain areas. Pcdh8 EE resulted in the reorganization of multiple TF expression crucial for correct DV patterning (e.g. Shh, Dbx1, Lhx2), suggesting reciprocal regulation between adhesion molecules and transcriptional networks.”*

3. Line 141-145: The intent of this section is unclear. Why was it necessary to examine embryos seven days after IUE? The significance of this timeframe is not well communicated.

Suggestion: Rewrite this section to clearly articulate the purpose of the seven-day observation and its relevance to the study's objectives.

We have clarified the rationale of the revised manuscript, which was to determine whether the aggregates persist beyond 48 hours post-electroporation and to assess whether the ectopic neurons induced by Dbx1 acquire a defined identity that is only visible at later stages, survive, and, if so, what specific cell identity they acquire. We have added the following to the Results section (lines 131-132): *“To determine whether Dbx1-induced neurons persist and acquire a defined identity at later stages, we analysed embryos seven days after IUE (E11.5-18.5)”*.

4. Line 141-145 (CLA Neuron Markers): The manuscript states that $Nr4a2^+$, $Ctip2^-$, and $Tle4^-$ cells are characteristic of CLA neurons, but no references are provided to support this claim.

Suggestion: Cite relevant literature to substantiate the claim that these markers define CLA neurons.

We would like to clarify that our intention was not to define $Nr4a2^+$, $Ctip2^-$ and $Tle4^-$ cells as characteristic of CLA neurons. We have provided several references of recent studies reporting CLA neuron identity at late embryonic/postnatal ages. In addition, we have removed all “data not shown”, including the Tle4 data. Finally, we have revised our interpretation of these cells as having a CLA-like identity in our study.

5. Line 60-65: The description suggests a direct link between aggregation and the conversion to CLA neuron fate, which may be an overly strong assertion.

Suggestion: Move this discussion to the Discussion section to frame it as a hypothesis or interpretation rather than a definitive result, and provide a more nuanced explanation.

We have revised this description to provide a more nuanced link (lines 150-159): *“Altogether, these results show that Dbx1 EE promotes cell-cell adhesion in a cell-autonomous manner within the LP/VP, leading to the induction of Nr4a2⁺Ctip2⁻ cells resembling CLA-like neurons. In addition, Dbx1-induced cell aggregation is spatio-temporally restricted by the competence of the cortical neuroepithelium, with no aggregation in DP (Arai et al., 2019), correlating with Dbx1 EE promoting exclusively Cajal-Retzius and subplate-like fates at E11.5 (Arai et al., 2019), and aggregation in LP/VP with CLA-like fate induction. Moreover, we confirm that endogenous VP Dbx1⁺ progenitors do not generate subplate neurons in mice, and subplate-like fate arises from primate-like Dbx1 expression in postmitotic neurons (Arai et al., 2019), whereas Cajal-Retzius and CLA-like neurons are endogenous fates driven by VP Dbx1⁺ progenitors in mice.”*

Also, we have added in the Discussion section of the revised manuscript (lines 384-391): *“Importantly, δ -Pcdhs mediate strictly homophilic adhesion and avoid heterotypic interactions (Bisogni et al., 2018). Even subtle mismatches in δ -Pcdh expression can prevent co-aggregation, implying that adhesive affinity and surface composition determine segregation behaviour. This principle likely explains why Dbx1 EE in the LP leads transfected cells into Pcdh8-expressing aggregates with CLA-like neuron profile (Nr4a2⁺Ctip2⁻Bhlhe22⁻Reln⁻Calb2⁻), whereas in the DP, it did not induce aggregation but instead correlated with Cajal-Retzius and subplate-like neuron fates (Arai et al., 2019).”*

6. Fig. 2A and 2D: The terminology used for brain regions (e.g., lateral anterior, lateral posterior) in Fig. 2A differs from the MP, DP, and LP nomenclature used in Fig. 1, which is confusing. Similarly, the regional divisions in Fig. 2D slightly differ from those in Fig. 1, potentially causing reader confusion. Suggestion: Add annotations in Fig. 2A and 2D (e.g., “corresponds to DP in Fig. 1”) to clarify the relationship with Fig. 1’s nomenclature and reduce confusion.

We thank the reviewer for this suggestion. We have updated the nomenclature in revised Fig. 2A to match the DP, MP, and LP labels used in Fig. 1C. The regional divisions in revised Fig. 2D differ slightly from those in revised Fig. 1C because revised Fig. 2D shows the proliferative domains, while revised Fig. 1C depicts the postmitotic compartments. We hope these changes have improved the clarity and interpretability of the figures.

7. Line 183-184: The sentence “We first used a previously generated dataset sampling cell diversity around the pallial-subpallial boundary (PSB) at E12.5” appears to be missing a reference to Fig. 2D. Suggestion: Add “(Fig. 2D)” after this sentence to ensure proper referencing.

We have included the missing reference in both the text and the figure legend.

8. Line 228-277: The latter half of this section (starting around Line 259) contains interpretive content that is more suitable for the Discussion section. The Results section should focus on presenting factual findings.

Suggestion: Move the interpretive content from Line 259 onward to the Discussion section to maintain clarity in the Results.

Similarly to comment 5, we have moved the interpretative content (lines 269-275 of the original manuscript) to the Discussion section of the revised manuscript and also revised it (lines 391-400): *“We propose that high Pcdh8 expression in Dbx1 EE-expressing cells, combined with low Pcdh19 levels preferentially in LP/VP progenitors, promoted strong homophilic interactions and aggregate formation in vivo. Conversely, higher co-expression of other δ -Pcdhs (i.e. Pcdh19) in the DP likely reduces aggregation due to heterotypic incompatibility. Together, we believe that the strong cell-segregative character of Pcdh8, combined with dynamic developmental changes in adhesion molecule combinatorial codes, regional (medial-dorsal-lateral axis) and progenitor-specific differences along the DV axis (i.e. VP and septum versus subpallium), modulate*

adhesion strength (Bisogni et al., 2018) and possibly neuronal migration, thereby contributing to the segregation of distinct neuronal populations”.

9. Line 307-308: The manuscript states that Pcdh8 overexpression causes cortical thickening, but Fig. 5A suggests that this thickening is specific to the electroporated region.

Suggestion: Clarify whether the thickening is specific to the electroporated area and, if so, explicitly state this in the text.

We thank the reviewer for pointing this out. Indeed, the thickening occurs specifically in the electroporated cortical region. The correct statement has been included both in the main text (lines 290-291): *“Upon Pcdh8 EE, the electroporated cortical region exhibited increased thickness compared to the contralateral non-electroporated hemisphere (Fig. 5A-C)”* and in the figure legend.

4. Discussion

1. Insufficient Mechanistic Depth: The molecular mechanisms by which Pcdh8 regulates Dbx1 expression (e.g., signaling pathways, post-transcriptional regulation) are not sufficiently discussed. For instance, the potential interaction of Pcdh8's intracellular domain (ICD) with specific kinases or transcriptional co-factors requires further exploration.

Suggestion: Based on existing literature, propose specific hypotheses regarding Pcdh8's ICD-mediated signaling and suggest future experimental designs to test these hypotheses.

We thank the reviewer for this insightful comment, which raises a very interesting point for further exploration. The intracellular mechanisms underlying the action of most Pcdhs are so far poorly characterized, so our elaboration is purely speculative.

Recent studies have shown that the intracellular domain (ICD) of Pcdh8, upon cis-binding to N-cadherin, activates the TAO2 kinase and p38-mitogen-activated protein kinase (MAPK). The TAO2 kinase has been shown to be a constituent and key player in nuclear speckle structure and function, including RNA splicing and nuclear export (Gao et al., 2022). MAPK can regulate, for example, tumor necrosis factor alpha (TNF-alpha) gene expression at a post-transcriptional level by controlling mRNA, protein expression and RNA-binding activity of the zinc finger protein tristetraproline (Mahtani et al., 2001). Furthermore, Pcdhs have been implicated in the regulation of both non-canonical (Unterseher et al., 2004) and canonical Wnt/ β -catenin, and AKT/GSK3B/ β -catenin pathways (Zong et al., 2017). Interestingly, GSK3 is engaged in nearly every aspect of neuronal development and function (Gizak et al., 2020). GSK3 activity can regulate, for example, Cadherin-11 expression in two ways: (1) a β -catenin-independent regulation of Cadherin-11 steady state mRNA levels, and (2) a β -catenin-dependent effect on Cadherin-11 3'UTR stability and protein translation (Farina et al., 2009).

Pcdh8 is one of the best-characterized Pcdhs linking cadherin-mediated adhesion to Wnt signaling. During *Xenopus laevis* gastrulation, Pcdh8 activates the Wnt/planar cell polarity (PCP) pathway by binding Frizzled7, coordinating cell polarity, and regulating convergent extension movements and tissue separation/morphogenesis (Medina et al., 2004; Unterseher et al., 2004; Kraft et al., 2012). Loss of Pcdh8 disrupts JNK activation via Rac1, confirming its essential role in Wnt/PCP signaling initiation (Unterseher et al., 2004). The ICD of Pcdh8 further modulates this pathway through interactions with Sprouty, *Xenopus* ANR5, Nemo-like kinase 1 (NLK1), and casein kinase 2B (CK2B), which influence PCP signaling strength, Pcdh8 stability, and canonical Wnt/ β -catenin activity (Chung et al., 2007, Wang et al., 2008, Kietzmann et al., 2012, Kumar et al., 2017). Interestingly, Pcdh8 expression itself is regulated by Wnt/PCP signaling, forming a feedback loop (Schambony and Wedlich, 2007).

Although the exact mechanism of Pcdh8's post-transcriptional action on Dbx1 expression remains unknown, the current literature thus suggests that it may be through its interaction with the TAO2, p38- MAPK or GSK3/ β -catenin post-transcriptional control of gene expression. That is what we have briefly discussed in the original manuscript (lines 426-430 and 469-474).

Following the reviewer's suggestion, we have expanded this section (lines 461-475) to further discuss the potential involvement of the Pcdh8's ICD in multiple signalling pathways relevant for regulating Dbx1 expression: *“Mechanistically, Pcdh8 may regulate TF expression, notably Dbx1, through post-transcriptional mechanisms mediated by its intracellular domain (ICD). The ICD links cadherin-mediated adhesion to Wnt signalling, crucial for cell fate, migration and tissue patterning (Pancho et al., 2020), by activating TAO2B and p38-MAP kinases, which regulate*

nuclear speckle function, RNA processing, and post-transcriptional gene expression (Gao et al., 2022; Mahtani et al., 2001; Yasuda et al., 2007). Pcdh8 also modulates both non-canonical and canonical Wnt/ β -catenin, and AKT/GSK3B/ β -catenin pathways (Unterseher et al., 2004; Zong et al., 2017). GSK3, a central regulator of neuronal development (Gizak et al., 2020), controls cadherin mRNA stability through β -catenin-dependent and -independent mechanisms (Farina et al., 2009). Moreover, Pcdh8 activates Wnt/planar cell polarity (PCP) signalling, coordinating cell polarity and morphogenesis (Medina et al., 2004; Kraft et al., 2012). Its ICD further interacts with Sprouty, NLK1, and CK2B signalling factors to fine-tune Wnt signalling and Pcdh8 stability, while Pcdh8 expression itself is regulated by Wnt/PCP activity, forming a feedback loop (Schambony and Wedlich, 2007)”.

2. Lack of Discussion on Limitations: The manuscript does not adequately address experimental design limitations, such as the regional specificity of IUE or potential off-target effects of Pcdh8 knockdown (KD). This makes it unclear how confident we can be in the interpretation of the results.

Suggestion: Explicitly describe the limitations of the experimental design and their potential impact on result interpretation.

Indeed, although we employed a validated shRNA sequence for Pcdh8 knockdown, potential off-target effects cannot be completely excluded. We have now acknowledge this (lines 510-511), along with other limitations raised by Reviewer 1, in the “Study limitations” section of the revised manuscript: *“Furthermore, studies using ectopic gene expression or knockdown strategies bear intrinsic drawbacks, such as non-physiological protein levels or potential off-target effects”.*

3. Clinical Relevance: The manuscript mentions that Pcdh8 mutations are associated with neurodevelopmental disorders like autism and schizophrenia, but it lacks a detailed discussion of how these findings relate to the current study.

Suggestion: Discuss how the results (e.g., CLA neuron abnormalities) might contribute to understanding behavioral deficits in neurodevelopmental disorders and their clinical significance.

We agree that discussing the potential implications of Pcdhs and their regulatory mechanisms in pathological conditions is of significant interest. However, due to strict space constraints, we have chosen to omit the discussion on neurodevelopmental disorders, as it is less directly relevant to the data presented in this study. If the reviewer or editor feels strongly against this decision, we would be pleased to reintroduce a brief section addressing potential clinical implications, such as:

“Given CLA’s critical involvement, including Nr4a2⁺ neurons, in attention control, slow wave sleep, depressive-like behavior, and pain processing (Puelles et al., 2016b; Bruguier et al., 2020; Mantas et al., 2024; Yan et al., 2025), our findings provide key insights into cortical development and functional organization. On the other hand, converging evidence defining the claustrum as a network hub for attention and salience with direct amygdala interplay (Rodríguez-Vidal et al., 2024), together with structural and functional findings in autism and schizophrenia (Schinz et al., 2025; Schumann et al., 2011), supports a role for the CLA circuitry in neurodevelopmental pathology. Moreover, as δ -Pcdh gene mutations have been associated to several neurodevelopmental disorders (Kahr et al., 2013), the Dbx1-Pcdh8 axis may offer a developmental entry point into the origins of circuit-level dysfunction in autism and schizophrenia.”

4. Data Interpretation: As noted in the Results section, the reason why Dbx1 EE-induced cell aggregation is specific to LP/VP is not sufficiently explained. The roles of other protocadherins (e.g., Pcdh9, Pcdh19) and regional expression pattern differences need further discussion.

Suggestion: Expand the discussion on how Pcdh8 and other protocadherins contribute to regional specificity, referencing Fig. S5 if applicable.

Please refer to our answer to comment 2 regarding Fig.1. We have revised what was already discussed in the original manuscript and have expanded on this topic in the Discussion section (lines 383-403), as it was also raised by reviewer 1: *“Consistently, our expression analyses show that δ -Pcdh (Pcdh8, Pcdh9, Pcdh19) display sharply distinct and dynamic spatial patterns*

between E11.5 and E13.5. Importantly, δ -Pcdhs mediate strictly homophilic adhesion and avoid heterotypic interactions (Bisogni et al., 2018). Even subtle mismatches in δ -Pcdh expression can prevent co-aggregation, implying that adhesive affinity and surface composition determine segregation behaviour. This principle likely explains why Dbx1 EE in the LP leads transfected cells into Pcdh8-expressing aggregates with CLA- like neuron profile (Nr4a2⁺Ctip2⁻Bhlhe22⁻Reln⁻Calb2⁻), whereas in the DP, it did not induce aggregation but instead correlated with Cajal-Retzius and subplate-like neuron fates (Arai et al., 2019). We propose that high Pcdh8 expression in Dbx1 EE-expressing cells, combined with low Pcdh19 levels preferentially in LP/VP progenitors, promoted strong homophilic interactions and aggregate formation in vivo. Conversely, higher co-expression of other δ -Pcdhs (i.e. Pcdh19) in the DP likely reduces aggregation due to heterotypic incompatibility. Together, we believe that the strong cell-segregative character of Pcdh8, combined with dynamic developmental changes in adhesion molecule combinatorial codes, regional (medial-dorsal-lateral axis) and progenitor-specific differences along the DV axis (i.e. VP and septum versus subpallium), modulate adhesion strength (Bisogni et al., 2018) and possibly neuronal migration, thereby contributing to the segregation of distinct neuronal populations. Moreover, variations in adhesion properties can play a crucial role in organising properly functioning brain areas. Pcdh8 EE resulted in the reorganization of multiple TF expression crucial for correct DV patterning (e.g. Shh, Dbx1, Lhx2), suggesting reciprocal regulation between adhesion molecules and transcriptional networks”.

5. Control Experiments: The specificity of Pcdh8 KD and EE experiments is supported by limited control experiments (e.g., non-specific shRNA or EE of other protocadherins). The claim that Pcdh8 KD suppresses Dbx1 expression via post-transcriptional regulation (page 11, Fig. 4) lacks supporting evidence from additional experiments (e.g., mRNA stability analysis or proteomics). Suggestion: Include control experiments with non-specific shRNA or other protocadherin KD in supplementary materials to confirm Pcdh8 KD specificity. Additionally, propose or include experiments to validate the post-transcriptional regulation mechanism.

We have used a scrambled shRNA in our experiments, which is a standard control procedure. We believe that additional testing would require new experiments, including the use of more animals and obtaining the necessary licenses.

Regarding the post-transcriptional mechanism, please refer to our previous answer to comment 1 of this reviewer about “Insufficient Mechanistic Depth”. The observation that GFP is expressed in transfected cells indicates that the sequence has been successfully transcribed and translated, and since Dbx1 EE is driven by a plasmid bearing a heterologous CAG promoter, the regulatory mechanism of Pcdh8 shRNA on Dbx1 expression is most likely post-transcriptional. We have included the clarification of our interpretation in the Discussion of the revised manuscript (lines 427-431): “While Dbx1 may control Pcdh8 expression at the transcriptional level, the use of the same pCAGGS promoter in both EE and KD experiments, together with GFP expression confirming successful transcription and translation, suggests that Pcdh8, in turn, modulates Dbx1 post-transcriptionally.”

5. Methods

1. Statistical Analysis Details:

- The normalization method for qPCR is unclear. The reference gene used for qPCR normalization (e.g., GAPDH or β -actin) should be specified.

Indeed, although the GAPDH qPCR sequence is provided in the Materials and Methods section, the description of the protocol and the normalization procedure was missing. We apologize for this omission. We have clarified these details in the Materials and Methods section (lines 771-773): “Expression data were normalized to the geometric mean of housekeeping gene (Gapdh), which showed stable expression across all samples”, as well as in the corresponding figure legends.

- The sample size details for IUE data analysis are not clearly described. For example, in Fig. 1E, it is stated that n=6, but it is unclear whether this represents six independent mother mice with one embryo sampled per mother or, for instance, two IUE experiments with three embryos sampled from each mother.

Suggestion: Specify the reference gene used for qPCR normalization and its rationale. Clearly state the number of mother mice and embryos used in IUE experiments for each Fig.

We have revised all figure legends to clarify this information for each experiment.

2. Knockdown Experiment Methods:

- The selection process for shRNA target sequences is not described. The target sequences and their selection criteria should be detailed.
- It is unclear whether non-specific shRNA controls were used to confirm the specificity and efficiency of Pcdh8 KD. Additionally, information on the extent of protein-level reduction (e.g., via Western blot) is needed.

Suggestion: Provide details on the shRNA target sequences and their selection process. Include data from non-specific shRNA controls and Western blot results showing the extent of Pcdh8 protein reduction in supplementary materials.

We believe the reviewer may have overlooked the information regarding the shRNA and control (scrambled) shRNA sequences, which are provided in Table 1 (page 27 of the original manuscript), as well as the method for selecting the most efficient shRNA (out of the three tested), described in lines 282-285 of the original manuscript. Nonetheless, we have also included this information in the Methods section (lines 723-725) to improve its visibility: *“The most efficient shRNA, reducing Pcdh8 expression by 51% in HEK293T cells 48h after co-transfection with the Pcdh8 expression vector, was identified by western blot analysis”*.

6. Other Minor Comments

* Line 194: The reference to Fig. 2D appears to be a mistake and should likely be Fig. 2E.

Suggestion: Correct the reference to Fig. 2E.

We thank the reviewer and have corrected the reference as indicated by the reviewer.

* Line 391: Adding "in the LP" after "E12.5" would improve clarity. Suggestion: Insert "in the LP" to specify the region.

We have corrected the sentence as indicated by the reviewer.

* Grammar: Some grammatical errors and unclear expressions are present, particularly in the latter half of the Discussion section, which feels verbose.

Suggestion: Have the manuscript proofread by a native English speaker to improve clarity and conciseness.

We have engaged a native scientific English speaker to revise our manuscript and address potential grammatical errors.

* Ethical Considerations: The description of ethical considerations for animal experiments is appropriate but lacks specific details, such as the ethics committee approval number or reference to specific guidelines.

Suggestion: Include the ethics committee approval number and reference to relevant guidelines to enhance transparency.

We believe the reviewer may have overlooked the information regarding ethical approval, which is provided in the Materials and Methods, Animals section, lines 702-704 (new lines 711-714).

Reviewer #3

Reviewer 3: SUMMARY OF THE ADVANCE MADE IN THIS PAPER AND ITS POTENTIAL SIGNIFICANCE TO THE FIELD

Pierani *et al.* describe a bidirectional interaction between PCDH8 and DBX1 that regulates neural identity and downstream developmental processes. This is an exciting and significant finding as

even if the regulation of cell adhesion molecules by transcription factors is well established; the opposite scenario, where the adhesion identities regulate specific expression of transcription factors has not been previously explored. This observation opens a new avenue in the field to enhance our understanding on the role that cell adhesion molecules play in determining cell identity.

We are pleased that the reviewer finds our manuscript interesting and recognizes its potential to open new avenues in the study of adhesion molecule function in brain development. Below we have addressed all of the reviewer's comments.

SUGGESTIONS TO AUTHORS

This is a well-written manuscript with clear and well-presented Figs. The experimental approach is rigorous, and the conclusions are supported by the data presented. There are only few considerations that will enhance the readability and the clarity of the manuscript:

1. In general, the text will benefit from less abbreviations, as it becomes hard to follow. In particular, authors should consider not abbreviating cell autonomous as CA and non-cell autonomous as NCA, it is not necessary, and it really decreases readability. Authors should also consider removing other abbreviations like CR for Cajal-Retzius, CLA for claustroramygdaloid complex, etc. Some abbreviations are used 1-2 times, so they are not needed, like ncPCDHs, and finally some abbreviations are never spelled out, like RC (rostror-caudal?).

We have reorganized the abbreviations and retained only those essential for the flow of the manuscript and for reducing the word count.

2. Authors should consider rearranging Figs as the panels do not follow the order by which they are referred to in the text. To enhance clarity, Figs should be numbered the way they appear (e.g. Fig. S5G- I is mentioned in the text before Fig. S3 and S4; Figs3C is mentioned before Fig. S3A), even the panels in the main Figs are not mentioned in the text in order (Fig. 2G is mentioned before Fig. 2D,E,F).

We have reorganized selected panels in the supplementary figures and adjusted the main text in the Results section to improve the overall flow and clarity of the manuscript.

3. Authors should include "data not shown" if it was used to draw conclusions (eg. "together with data showing Nr4a2⁺ Dbx1-derived neurons in a E14.5 dataset using Dbx1Cre;TauLacZ reporter (data not shown), these findings indicate that Dbx1 progenitors in the mouse VP originate pallial Nr4a2⁺ neurons that migrate ventrally, corresponding to neurons of the future claustroramygdaloid complex (CLA) (Puelles *et al.*, 2016a), and not SP neurons.")

We have decided to remove these data from the manuscript, as the results are supported by other findings within the submitted study (histology showing Dbx1-derived cells expressing Nr4a2, revised Fig. S3) as well as by previously published data from Moreau *et al.*, 2021.

4. Please provide a more comprehensive rationale and details about how cells were delineated in panels a-d of Fig. 2H. This is very relevant as is the evidence provided to claim pcdh8 and dbx1 co-expression.

We appreciate the reviewer's concern regarding the accurate indication of Pcdh8 and Dbx1 co-expression in the same cells in Fig. 2H. To improve clarity, we have revised the dashed circles in revised Fig. 2H, highlighting only cells in which co-localization of *Pcdh8* and *Dbx1* is clearly visible at the presented magnification. Additionally, we have added arrowheads pointing to *Pcdh8* and *Dbx1* signals within the same cell. To further enhance transparency and reproducibility, we have included the following clarification in the Materials and Methods section under FISH (lines 792-793): "Co-localization of *Pcdh8* (red signal) and *Dbx1* (green signal) was identified in cells outlined by DAPI-stained nuclei."

5. In Fig. 3D, there's no housekeeping gene for normalization, authors should include such gene to make sure the quantity/quality of the mRNA is comparable between groups/samples.

Indeed, as also pointed out by Reviewer 2, although the *Gapdh* qPCR sequence is provided in the Materials and Methods section, the description of the protocol and the normalization procedure was missing. We have clarified these details in the Materials and Methods section (lines 771-773): “Expression data were normalized to the geometric mean of housekeeping gene *Gapdh*, which showed stable expression across all samples”, as well as in the corresponding figure legends.

6. Graph in Fig. 4F is missing the Dbx1EE + control shRNA bar, this is particularly important as authors observed numerous GFP⁺Dbx1⁻ cells in comparison to Dbx1 EE alone. This observation was explained by the fact that the plasmid used for Dbx1 EE is driven by a heterologous CAG promoter and thus Pcdh8 shRNA likely affects Dbx1 expression post-transcriptionally. If this is the case, then the control shRNA should have the same effect.

We have now provided the quantification of Dbx1-positive cells following electroporation with the combination of Dbx1 EE + control shRNA in revised Fig. 4F, showing an increased percentage of Dbx1⁺ cells among the total GFP⁺ cells, similar to that observed with Dbx1 EE alone.

Minor

- Please replace insets c,d in Fig. 1I for better quality pictures.

We believe the reviewer is referring to potential overexposure of the provided images, as also noted by Reviewer 1. We have replaced these images with panels that have been properly adjusted.

- For Fig. 5G is hard to see the lamination in the control and thus to observe the described delamination in the PCDH8 EE, please replace the pictures or add arrows to orient the reader.

The delamination from the VZ is already visible in the close-up (panel a), where PH3-positive cells (normally located at the ventricular lumen) are trapped within rosette-like structures, supporting and indicating delamination from the VZ. To improve readability, we have added dashed lines delineating the VZ and empty arrowheads in the Control image indicating PH3-positive cells aligned with the VZ, contrasting with the delaminated dividing cells (white arrowheads) found in Pcdh8 EE.

- Fig. 7 description should have its own title and be separated from the previous section.

We have separated the section on Notch signaling from the rescue experiment, as suggested by the reviewer, providing the new title “*Pcdh8 EE controls progenitor identity by impairing Jag1-mediated Notch signaling*” (line 329).

Second decision letter

MS ID#: dev.205011R1

MS TITLE: Bidirectional interaction between Protocadherin 8 and transcription factor Dbx1 regulates cerebral cortex development

AUTHORS: Alessandra Pierani, Andrzej W. Cwetsch, Sofia Ferreira, Elodie Delberghe, Javier Gilabert-Juan, Matthieu X. Moreau, Yoann Saillour, Pau Garcia Bolufer, Saray Calvo Parra, Jose González Martínez, Durcia Massoukou, Ugo Borello and Frédéric Causeret

Dear Dr Pierani,

I am happy to tell you that your manuscript has been accepted for publication in Development, pending our standard publication integrity checks.

Reviewer 1

Advance summary and potential significance to field

I appreciate the time and care the authors have spent in addressing the comments. The revisions have strengthened the manuscript, and I recommend it for publication.

Reviewer 2

Advance summary and potential significance to field

The authors have fully and clearly addressed all comments.

Comments for the author

I recommend acceptance.

Reviewer 3

Advance summary and potential significance to field

Authors addressed all concerns from this reviewer.